# Blood–Brain Barrier Conquest in Glioblastoma Nanomedicine: Strategies, Clinical Advances, and Emerging Challenges

**DOI:** 10.3390/cancers16193300

**Published:** 2024-09-27

**Authors:** Mengyun Duan, Ruina Cao, Yuan Yang, Xiaoguang Chen, Lian Liu, Boxu Ren, Lingzhi Wang, Boon-Cher Goh

**Affiliations:** 1Department of Medical Imaging, Health Science Center, Yangtze University, Jingzhou 434023, China; mengyun-duan@yangtzeu.edu.cn (M.D.); chen_xg@yangtzeu.edu.cn (X.C.); 2Department of Anesthesiology, Maternal and Child Health Hospital of Hubei Province, Wuhan 430070, China; zijun_wu@whu.edu.cn; 3Department of Radiology, Renmin Hospital of Wuhan University, Wuhan 430060, China; rm003237@whu.edu.cn; 4Department of Pharmacology, Health Science Center, Yangtze University, Jingzhou 434023, China; liulian@yangtzeu.edu.cn; 5NUS Center for Cancer Research (N2CR), Yong Loo Lin School of Medicine, National University of Singapore, Singapore 117599, Singapore; phcgbc@nus.edu.sg; 6Department of Pharmacology, Yong Loo Lin School of Medicine, National University of Singapore, Singapore 117600, Singapore; 7Cancer Science Institute of Singapore, National University of Singapore, Singapore 117599, Singapore; 8Department of Haematology-Oncology, National University Cancer Institute, Singapore 119228, Singapore

**Keywords:** glioblastoma, blood–brain barrier, drug delivery, nanomedicine delivery system, nanoparticles

## Abstract

**Simple Summary:**

Glioblastoma (GBM), a serious brain cancer, has poor treatment outcomes despite surgery, radiation, and chemotherapy. The blood–brain barrier (BBB) makes GBM-targeted drug delivery difficult. Recent studies have shown that nanomedicine delivery systems (NDDSs) can target GBM safely and effectively. In this review, we look at ways to overcome the BBB with NDDSs in preclinical studies, summarize the clinical progress, and discuss strategies to improve NDDSs and speed up their use in GBM treatment through clinical trials.

**Abstract:**

Glioblastoma (GBM) is a prevalent type of malignancy within the central nervous system (CNS) that is associated with a poor prognosis. The standard treatment for GBM includes the surgical resection of the tumor, followed by radiotherapy and chemotherapy; yet, despite these interventions, overall treatment outcomes remain suboptimal. The blood–brain barrier (BBB), which plays a crucial role in maintaining the stability of brain tissue under normal physiological conditions of the CNS, also poses a significant obstacle to the effective delivery of therapeutic agents to GBMs. Recent preclinical studies have demonstrated that nanomedicine delivery systems (NDDSs) offer promising results, demonstrating both effective GBM targeting and safety, thereby presenting a potential solution for targeted drug delivery. In this review, we first explore the various strategies employed in preclinical studies to overcome the BBB for drug delivery. Subsequently, the results of the clinical translation of NDDSs are summarized, highlighting the progress made. Finally, we discuss potential strategies for advancing the development of NDDSs and accelerating their translational research through well-designed clinical trials in GBM therapy.

## 1. Introduction

Glioblastoma (GBM) is the most prevalent malignant primary brain tumor (accounting for 13% of all tumors and 50.1% of all malignant tumors), with a 5-year survival rate of less than 7% [1,2]. Despite progress in multimodal treatments encompassing surgery, radiotherapy, chemotherapy, and targeted therapy, the overall prognosis remains unfavorable, with a median survival duration of merely 8 months [1]. Figure 1 illustrates the seminal milestones in GBM research and treatment. In a 2021 study, GBM, classified as IDH-wildtype and defined as WHO grade 4, specifically refers to diffusely infiltrative gliomas exhibiting classic histopathological features or molecular alterations [3]. Owing to its heterogeneity, aggressiveness, and the difficulty in distinguishing the malignant tissue margins compared to other solid tumors, along with the existence of the blood–brain barrier (BBB), up to 70% of patients undergo rapid disease deterioration within one year of diagnosis. Chemotherapy is one of the standardized steps in GBM treatment [4]. However, non-targeted administration often leads to low efficacy and systemic toxicity, while the presence of the BBB demands higher dosages of chemotherapeutic drugs. Developing brain-targeting drugs or formulations that can cross this barrier is urgently needed for effective treatment while minimizing toxic side effects.

The BBB is a natural defense mechanism between the bloodstream and brain tissue. An intact BBB effectively segregates brain tissue from peripheral blood and plays a pivotal biological role in upholding the fundamental stability of the internal brain tissue environment and the normal physiological state of the central nervous system (CNS) (Figure 2a). The BBB restricts nearly all large molecules and over 98% of small-molecule candidate drugs from permeating into it [5,6,7]. Therefore, the key issue in GBM drug therapy is the design and development of targeted nanomedicine delivery systems (NDDSs) that can effectively cross the BBB and ensure that drugs reach their intended sites in the brain. The two main approaches to delivering chemotherapy drugs to GBMs are crossing and bypassing the BBB (Figure 2b).

Biomaterials, encompassing both natural and synthetic sources, are materials that possess excellent biocompatibility and physicochemical properties and that are utilized in drug delivery. These materials can either be directly conjugated with drugs to enhance their pharmacokinetic properties or serve as excipients, leveraging nanotechnology to encapsulate drugs within nanocarriers (10–500 nm) or attach/adsorb them onto the carrier surface, thereby forming NDDSs that achieve the sustained, controlled release and targeted delivery of drugs [8]. The development of NDDSs represents a promising avenue for GBM treatment, although it is not without its challenges. A significant bottleneck lies in the transition from preclinical research to clinical trials. The heterogeneity of GBM presents a challenge, as it may lead to varied therapeutic responses. Moreover, the BBB further complicates drug delivery. Despite notable advancements made by NDDSs in overcoming the BBB, translating these preclinical successes into clinical practice has been hindered by issues related to scalability, regulatory pathways, and clinical validation.

Recent reviews have delved into the application of nanomedicine in brain tumor pharmacotherapy, highlighting the significance of nanotechnology in overcoming the BBB [9,10,11]. Nevertheless, amidst the heterogeneity of GBM and the constraints imposed by the BBB on drug delivery to tumor sites, the clinical advancements in nanodrug delivery systems remain largely uncharted territory. In this review, we aim to systematically classify the strategies for crossing and bypassing the BBB, highlight the key research accomplishments from the past 5 years using these systems, and discuss their strengths and weaknesses. Finally, we offer our perspectives on the future developments and challenges in this field, aiming to provide more comprehensive and in-depth guidance for the practical application of nanomedicine in GBM treatment.

## 2. NDDS Strategies for Crossing the BBB

The BBB is a formidable obstacle for drug delivery to the brain due to its tightly regulated selective permeability. However, several strategies have been developed to enhance the delivery of nanomedicines across the BBB.

### 2.1. Receptor-Mediated Transport

Receptor-mediated transport (RMT) is a pivotal approach in NDDSs for crossing the BBB, as it exploits the natural transport mechanisms of the brain’s protective interface. The existence of nutrient transport receptors on BBB endothelial cell membranes is crucial for sustaining normal brain functioning in humans, with some of them overexpressed on GBM cell membranes. A tailored nanomedicine delivery system enhances the BBB permeability and GBM cell targeting by binding to these receptors (Figure 3) [12].

#### 2.1.1. Transferrin and Lactoferrin Receptors

The overexpression of transferrin (Tf) and lactoferrin (Lf) receptors on BBB endothelial cells and glioma cells presents an opportunity for targeted drug delivery. Tf-functionalized nanoparticles (Tf-NPs) cross the BBB and target GBMs in intracranial orthotopic models, reducing the tumor burden and extending survival [13]. Lf, an iron-binding glycoprotein, surpasses Tf and OX-26 in BBB permeation [14]. Dong et al. developed a glioma-targeted drug delivery system utilizing biodegradable periodic mesoporous organosilica nanoparticles modified with Lf ligands, enhancing drug delivery to brain gliomas [15]. Taskeen et al. introduced an Lf receptor-mediated nanomedicine, USLP-NH_2_-PEG-TMZ-Lf, with enhanced BBB permeability and cytotoxicity for GBM treatment [16].

#### 2.1.2. Acetylcholine Receptors

Nicotinic acetylcholine receptors are widely distributed on the surface of the BBB, serving as a crucial target for drug delivery. 2-methacryloyloxyethyl phosphorylcholine (MPC)-based protein nanocapsules enhance CNS protein transport. Chen et al. engineered an NDDS containing TMZ (TMZ@RVG-Zein NPs), penetrating the BBB via receptor-mediated endocytosis. This NDDS exhibits exceptional biocompatibility and is capable of penetrating into GBM (U87) cell lines, thereby facilitating the release of TMZ for therapeutic effect [17].

#### 2.1.3. Folate Receptors

BBB and glioma cells express folate receptors (FRs), which are crucial for tumor DNA replication. Folate-modified NPs are internalized by tumor FRs via receptor-mediated endocytosis [18]. Xiang Yang Zhong et al. engineered folate/iRGD-modified NPs, enhancing targeting and uptake, delivering TMZ to nuclei, and inhibiting tumor proliferation. FRs can be integrated with other drugs for novel delivery systems (e.g., WGA/FA-MPEG-PCL NPs loaded with ETO, BCNU, or DOX) targeting the BBB to inhibit GBM growth [19].

#### 2.1.4. Low-Density Lipoprotein Receptors

Members of the LDL receptor superfamily exhibit a binding affinity towards a diverse range of ligands, including lipoproteins, proteases, and protease inhibitor complexes, facilitating their transport into the cell nucleus. Specifically, apolipoprotein E demonstrates selective binding to LDL receptors, while solid lipid nanoparticles functionalized with apolipoprotein E have been demonstrated to enhance BBB delivery by 1.5 times compared to non-functionalized nanoparticles [20]. Jiantang Liang and colleagues designed versatile biomimetic nanoplatform L-D-I/NPs to selectively target GBMs by binding to low-density lipoprotein receptacle-associated protein-1 (LRP1) and crossing the BBB, effectively inhibiting the progression of orthotopic GBM and significantly prolonging survival [21]. Angiopep-2-modified nanoparticles selectively target LRP1. Compared with the control group, the intravenous injection of Ti@FeAu-Ang nanoparticles resulted in a 10-fold reduction in tumor volume [22]. The Au-DOX@PO-ANG NPs designed by Chen He and colleagues also significantly reduced the GBM size in mice [23].

#### 2.1.5. Epidermal Growth Factor Receptors

The epidermal growth factor receptor (EGFR), an ErbB family tyrosine kinase receptor, governs GBM angiogenesis and determines high proliferation and drug resistance. EGFR-targeted drugs, such as cetuximab, panitumumab, nimotuzumab, and necitumumab, have been clinically approved [24]. EGFR overexpression occurs in 40–70% of GBM patients, with EGFRvIII, a cancer-specific deletion, present in 25–50% of pleomorphic GBMs [24]. The absence of EGFRvIII in normal tissues makes it a prime target for GBM-specific receptor therapies. The antitumor efficacy of panitumumab-conjugated and TMZ-loaded poly (lactic-co-glycolic acid) nanoparticles (PmAb-TMZ-PLGA-NPs) in GBM cells with overexpressed EGFRs is significantly enhanced through the inhibition of caspase-mediated autophagy via this novel NDDS, thereby promoting apoptotic cell death [25].

#### 2.1.6. Human Insulin-like Growth Factor-1 Receptors

The insulin-like growth factor-1 receptor (IGF1R), a receptor tyrosine kinase, is prominently expressed in brain endothelial cells, brain microvessels, and specific neuronal regions, and it mediates endocytosis upon high-affinity binding with IGF1, making it a suitable target for brain drug delivery [26,27]. High-affinity anti-IGF1R monoclonal antibodies penetrate the BBB after in situ brain perfusion (ISBP) administration [26]. Additionally, IGF1R5 effectively binds to IGF1R on the BBB, enhancing drug transport [27].

#### 2.1.7. Integrins

The αvβ3/αvβ5 integrins are upregulated in angiogenic sites and GBMs, underpinning the reliance on their BBB endothelial and GBM cell expression for GBM-targeted drug delivery systems [28]. Studies confirm that cyclic Arg-Gly-Asp (cRGD)-installed micelles enhance the targeted delivery and therapeutic efficacy of epirubicin in orthotopic GBM models [29]. Additionally, α5β1 integrin receptor overexpression in GBM cells facilitates the use of RGDk–lipid nanoparticles for the concurrent delivery of chemotherapeutics and siRNA, significantly inhibiting tumor growth in mouse models [30]. These discoveries underscore the efficacy of integrin-mediated carriers for drug delivery in aggressive GBM.

#### 2.1.8. CD13

CD13 is widely overexpressed on glioma neovasculature endothelial cells and glioma cell surfaces [31]. Asn-Gly-Asp (NGR), a structural motif for endothelial cells and the tumor neovasculature, functions as an effective tumor-targeting drug carrier. Circular iNGR in the blood can specifically and rapidly bind to CD13, demonstrating a strong tumor vascular-targeting ability [32]. Sai An et al. engineered PEGylated iNGR-modified RNAi nanoparticles that exhibit superior tumor accumulation and penetration, offering a novel avenue for targeted GBM therapy [31].

#### 2.1.9. Neuropilin-1

Neuropilin-1 (NRP-1), a transmembrane glycoprotein, is overexpressed in GBM cells and serves as a co-receptor for semaphorin3A and vascular endothelial growth factor, contributing significantly to tumor angiogenesis, growth, and metastasis [33]. CPT-S-S-PEG-iRGD@IR780 nanoparticles were modified via an iRGD peptide and IR780 photosensitizer, which can be used as a drug delivery system. Leveraging the αvβ integrin and NRP-1-mediated transport, this system efficiently traverses the BBB to target GBMs, enhancing the antitumor efficacy in combined therapies [34].

#### 2.1.10. Heat Shock Protein 70

Heat shock protein 70 (Hsp70) is selectively expressed on GBM cell membranes and serves as a precise target for GBM therapy, enhancing the binding efficiency [35]. Acid-triggered gold nanoparticles (D-A-DA/TPP) selectively deliver DOX to glioma tissue via Hsp70 targeting, with TPP (a peptide that binds to Hsp70 on the membranes of glioma cells) facilitating cellular uptake. Under the weakly acidic tumor microenvironment, D-A-DA/TPP aggregation prolongs circulation, augments binding, and triggers a pH-responsive DOX release [35]. Jianfen Zhou et al. devised pHA-AOHX-VAP-DOX, a nanodrug system that traverses the BBB via dopamine and GRP78 receptors, a heat shock protein family member, to extend the survival in nude mice with intracranial U87 gliomas [36].

These examples underscore the versatility and potential of RMT in nanomedicine, particularly for GBM therapy. By binding to specific receptors overexpressed on the BBB and tumor cells, nanoparticles can be guided to their target, enhancing drug delivery efficiency while minimizing systemic side effects. The continued exploration and refinement of RMT strategies hold great promise for improving the prognosis and treatment of brain-related diseases.

### 2.2. Transporter-Mediated Transport

Membrane transport proteins in neurovascular units, comprising 10–15% of their structure [37], are essential for the transmembrane movement of diverse substrates, including small water-soluble molecules. These proteins, categorized into four types based on their mechanisms, undergo conformational changes to facilitate transport and can be targeted by nanoparticles for drug delivery [38]. Their strategic distribution and unique mechanisms on brain capillary endothelial cells (BCECs) enhance the targeting and uptake of nanomedicines in brain tissue (Figure 4). Therefore, transporter-mediated transport (TMT) has become an important method in NDDSs for crossing the BBB.

#### 2.2.1. Glucose Transporters

Glucose is the brain’s primary energy source, and glucose transporter-1 (GLUT1) is its predominant transporter on BCEC membranes, facilitating its rapid translocation into brain tissue [40]. Leveraging post-fasting hyperglycemia, nanocarriers can enhance drug delivery across the BBB via GLUT1 [41]. Glycosylated derivatives of the heptapeptide ATWLPPR (A7R) have shown improved serum stability and enhanced BCEC uptake via GLUT1-mediated transcytosis, thereby improving BBB penetration and glioma cell absorption [42]. Zhang et al. developed a 2-DG nanocapsule system that exploits GLUT1 overexpression for targeted delivery to the GBM tumor microenvironment [39].

#### 2.2.2. Choline Transporters

The choline transporter (ChT) protein, prominently expressed on cerebral capillary endothelial cell lumens, efficiently traverses the BBB to transport acetylcholine and choline analogs into the CNS [43]. Li’s team engineered a cholinergic derivative-modified delivery system that significantly increased the glioma drug accumulation, inducing apoptosis and enhancing therapeutic outcomes [44]. MPC-n (IVIg) utilizes high-affinity ChT1 overexpression for targeted brain tissue drug delivery, reducing therapeutic doses [45]. Hairong Wang’s team devised a universal BBB-permeable smart polymer, pMPC-co-(anti-PD-L1-pPEGMA), that crosses the BBB via choline transporters to effectively treat malignant gliomas [46].

#### 2.2.3. Amino Acid Transporters

L-type amino acid transporter 1 (LAT1) is overexpressed in both the BBB and GBM cells [47], facilitating targeted brain drug delivery and minimizing peripheral exposure [48]. MeHg-L-cys harnesses LAT1 to enhance malignant glioma cell targeting and mitigate normal brain tissue toxicity [49]. Amphi-DOPA-loaded wp1066 nanocarriers significantly improved the survival in orthotopic mouse GBM models [50].

#### 2.2.4. Vitamin Transporters

The sodium-dependent vitamin C transporter (SVCT2), expressed in choroid plexus epithelial cells and brain tumor cell lines, emerges as a promising nanomedicine target [51]. The CNS penetration of ascorbic acid (AA), facilitated by its reversible oxidation to dehydroascorbic acid (DHAA), leverages the SVCT2-mediated transport mechanism, enhancing DHAA CNS levels and offering a strategy for drug delivery systems with potential in Alzheimer’s therapy [51,52]. Yao Peng’s research introduced a glucose–vitamin C-modified liposome for paclitaxel delivery to the brain, demonstrating superior targeting efficacy over unmodified or singly modified formulations [53].

#### 2.2.5. Organic Cation Transporters

Organic cation/carnitine transporter 2 (OCTN2, SLC22A5), a member of the OCTN family, is implicated in drug BBB permeation [54] and is upregulated in GBM [55]. Kou’s team engineered L-carnitine-conjugated nanoparticles (LC-PLGA NPs) that exploit the OCTN2 overexpression on brain endothelial and glioma cells for enhanced BBB permeability and targeted glioma cell internalization [56].

#### 2.2.6. Organic Anion Transporters

Organic anion-transporting polypeptides (OATPs) are a family of multi-specific transporters, including OATP1A2, that facilitate CNS drug uptake at the human BBB [57]. Statins illustrate this by crossing the BBB via the Oatp1a4 transporter, underscoring the utility of OATPs in targeted CNS drug delivery. Brain-specific anion transporter 1 (BSAT1), exclusively present in brain microvascular endothelial cells, is pivotal in targeted drug delivery within tumors and their surrounding areas. Research has demonstrated that platinum–nanogel conjugates with Cx43 and BSAT1 antibodies effectively shrink tumors [58]. Given the OATP substrate selectivity, the current research prioritizes stroke therapy, with glioma treatment often considered a secondary or dual-targeted approach.

#### 2.2.7. Monocarboxylate Transporters

Monocarboxylate transporter (MCT) family members mediate the cellular translocation of monocarboxylic acids, such as lactate and pyruvate, across various tissues. MCT1–4, integral to the plasma membrane, facilitates the bidirectional exchange of short-chain monocarboxylic acids and protons in mammalian cells [59]. Glioma cells uptake β-Hydroxy-β-methylbutyrate through H+-coupled MCTs [59], and the overexpression of MCT1 and MCT4 on their surfaces correlates with a poor prognosis, indicating their diagnostic and therapeutic potential [60]. Huber’s research demonstrated a link between MCTs and the brain penetration of cyclic C5-curcuminoids, emphasizing their role in BBB traversal [61].

### 2.3. Adsorptive-Mediated Transport

The nanocarrier size and surface charge are pivotal for BBB endocytosis, with cationic systems such as PEG enhancing drug solubility and cellular uptake through electrostatic interactions with the endothelial cell membrane [62]. The tumor blood vessel and cellular upregulation of negatively charged glycoproteins increases the nanocarrier accumulation in brain tumors (Figure 5) [63], enabling adsorptive-mediated transport (AMT) to serve as a method in NDDSs for crossing the BBB. While common transport receptors such as albumin and cell-penetrating peptides (CPPs) improve brain penetration [64], the lack of selectivity in electrostatic targeting limits its systemic drug administration use, necessitating combinations with specific ligands for enhanced targeting.

#### 2.3.1. Cationic Albumin

Electrostatic interactions between the cationic and anionic microdomains on BBB endothelial membranes initiate AMT, facilitating brain drug delivery, particularly for macromolecules [66]. Cationized albumin (pI > 8) accumulates in tumor cells via AMT [63] and can be integrated with anti-glioma drugs into a nanodrug delivery system to induce tumor cell death and retard growth [67]. Cationized immunoglobulin, monoclonal antibodies, and histones possess brain-targeting properties through similar mechanisms. Given AMT’s non-selectivity, it is often combined with other approaches to enhance the BBB permeation of NDDSs [68].

#### 2.3.2. Cell-Penetrating Peptides

CPPs encompass natural proteins (e.g., TAT, penetration peptides, Syn-B carriers) and synthetic, cationic, highly hydrophilic proteins such as polyarginine peptides [69]. Coupling CPPs with nanobodies enhances brain penetration [64]. TAT exhibits BBB-targeting potential, with its brain-targeting efficacy correlating positively with its bound ligand’s positive charge [70]. TAT-modified gold NPs outperform free doxorubicin in BBB traversal, GBM targeting, and circulation time [65]. However, the non-specific electrostatic membrane interactions of CPPs lack cell selectivity, elevating side effects. Researchers have devised dual/multi-ligand systems by conjugating CPPs with specific targeting ligands to optimize BBB traversal, impart cell selectivity, and enhance drug delivery efficiency [65,71].

### 2.4. Cell-Mediated Transport

The unique structures, mechanical properties, and surface ligands of human cells dictate their diverse physiological functions, fostering the development of cell-based, targeted drug delivery systems [72,73]. Cells serve as drug delivery vectors, encapsulating or attaching drugs and utilizing their homing mechanisms for efficient, targeted delivery [74], enabling cell-mediated transport (CMT) to serve as a potential method in NDDSs for crossing the BBB. Promising candidates include erythrocytes, leukocytes, and stem cells. Furthermore, synthetic nanodrug delivery systems utilize native cell membranes to enhance BBB penetration and active targeting, broadening the range of cells for drug delivery applications (Figure 6) [72].

#### 2.4.1. Erythrocytes

Erythrocytes have garnered attention as promising nanodrug carriers owing to their abundance, distinctive biconcave morphology, and prolonged circulation lifespan of 110–120 days [76]. Their structural characteristics, notably their biconcave shape and absence of a nucleus, enhance their drug encapsulation efficiency by up to 67% [77]. However, direct drug integration into or onto erythrocyte membranes can inflict irreparable damage, hastening clearance by the reticuloendothelial system and diminishing circulatory persistence [76]. To circumvent this, CPPs are utilized for drug loading, safeguarding the membrane integrity and function [78]. Despite these merits, erythrocytes as drug carriers encounter limitations such as uncontrolled drug release and the absence of specific targeting receptors. To address these limitations, red blood cell (RBC) NPs, nanoparticles coated with RBC membranes (RBCms), combine membrane functionalities with nanoparticle physicochemical properties, enhancing the loading capacity, stability, biocompatibility, and prolonged retention of drugs [79]. Furthermore, RBC-NPs exhibit refined BBB penetration and tumor-targeting abilities, facilitated by surface-modified ligands [80]. The novel nanodrug Ang-RBCm@NM-(Dox/Lex), functionalized with the surfaces of Angiopep-2-modified RBCms, exhibits a prolonged circulation time, superior BBB penetration, and enhanced tumor accumulation, effectively suppressing tumor growth and significantly extending the median survival time of orthotopic U87MG human GBM tumor-bearing nude mice [81]. Dong Luo and colleagues developed an NDDS using the tumor-penetrating peptide iRGD (CRGDK/RGPD/EC) derived from the RBC membrane as a carrier to overcome the BBB, enhance drug targeting, and increase the 30-day survival rate from 0% to 100% [82]. Mingming Song et al. devised LMP RFA NPs, a biomimetic nanodrug delivery system for targeted GBM therapy. This system encapsulates lomitapide (LMP)-loaded tetrahedral DNA nanocages within a folate-modified erythrocyte–cancer cell–macrophage hybrid membrane (FRUR) shell, demonstrating a high BBB permeability, precise tumor targeting, low side effects, and extended survival in tumor-bearing mice [83].

#### 2.4.2. Leukocytes

Leukocytes migrate to disease sites, traverse the BBB, and penetrate hypoxic tumor regions, facilitating targeted drug delivery to challenging areas [73]. These cells, including T cells, neutrophils, monocytes/macrophages, and dendritic cells, exhibit an innate affinity towards inflammation and are recruited to lesion sites via inflammatory factors [84].

Chimeric antigen receptor (CAR) T cells are genetically modified autologous T cells that can be employed for GBM-targeted drug delivery [85,86]. Despite the genetic modification of T cells, limited transport across the BBB remains one of the primary challenges encountered in CAR T-cell therapy. Focused ultrasound (FUS) was utilized to open the BBB and enhanced survival rates by 129% compared to CAR T-cell therapy alone [87]. Gloria B. Kim and her team developed a combined selective NDDS that utilizes high-affinity TQM-13 CAR T cells as drug carriers integrated with nanoparticles and DOX to enhance drug bioavailability while mitigating systemic toxicity [88]. Overall, T cells remain a promising delivery vehicle for GBM treatment.

Neutrophils (NEs), the predominant immune cells with rapid responsiveness to inflammatory stimuli, are recruited to lesion sites upon activation and can traverse the BBB, accessing inflamed brain tumor tissues [79,89]. A recent study harnessed this property by formulating an NE–exosome (NEs-Exos) system for GBM therapy, enabling the effective loading and intravenous delivery of DOX, which significantly suppressed tumor growth and prolonged survival [89]. Inspired by CAR T-cell therapy, researchers have engineered CAR neutrophils to specifically deliver tumor microenvironment-responsive nanodrugs to GBMs, combining chemotherapy with immunotherapy to minimize the off-target effects and extend the lifespan of tumor-bearing mice [90]. Utilizing the neutrophil membrane as a biomimetic drug delivery system, it interacts with adherent proteins to target endothelial cell membranes within the BBB for getting across [91].

Monocytes/macrophages, recruited to lesion sites by chemotactic factors [79], exhibit a robust migratory capacity towards GBMs, rendering them ideal carriers for cell-mediated brain drug delivery [92]. Macrophage-derived delivery systems are tumor-targeted, releasing therapeutics for efficacy [79]. Inflammatory macrophage membranes enhance targeting, and their integration with nanoparticles augments BBB traversal and brain targeting [93]. Xiao et al. achieved prolonged circulation, improved the BBB penetration, and enhanced the chemotherapeutic/chemodynamic efficacy for orthotopic C6 glioma in a mouse model by coating hybrid nanogels with macrophage membranes [75].

Dendritic cells, specialized antigen-presenting cells, stimulate and regulate innate and adaptive immune responses and are crucial for cytotoxic T-cell activation and proliferation. Dendritic cell membrane proteins facilitate BBB traversal. Xiaoyue Ma et al. devised aDCM@PLGA/RAPA, a nanoplatform coated with dendritic cell membranes, to effectively traverse the BBB, promoting the tumor immune response and synergistically augmenting GBM eradication in conjunction with rapamycin (RAPA) [94].

Despite their potent antitumor and immune regulatory capabilities, natural killer (NK) cells have been underutilized as carriers in research. NK cells inherently possess direct cytotoxic effects, acting as effective “antitumor agents” [73]. They can induce target cell apoptosis by releasing perforin and granzymes, which may cause the explosive release and hinder the penetration of tumor drugs.

#### 2.4.3. Stem Cells

Stem cells, such as mesenchymal stem cells (MSCs), neural stem cells (NSCs), and adipose-derived stem cells (ADSCs), are ideal carriers for glioma drug delivery due to their self-renewal, low immunogenicity, and easy differentiation. These stem cells can penetrate the BBB, migrate to tumors, and are used to treat hypoxia, necrosis, inflammatory tissue and gliomas, and even CNS diseases and other tumors [73,79]. MSCs and ADSCs, with their immunosuppressive properties, can be engineered to deliver therapeutic agents to tumors. The viral transfection of stem cells has been shown to induce immunoactive cytokine expression, thereby enhancing mouse survival rates [95]. ADSCs deliver apoptosation-inducing ligands (TRAILs) to brain tumors via viral transfection [96], and nonviral nanoparticles are also used to transfect stem cells targeting GBM in vitro [97]. The cancer-homing ability of ADSCs and the therapeutic potential of TRAILs can induce the apoptosis of primary tumor and microsatellite cells and prolong the survival time of GBM xenografts [97]. The transfection of ADSCs with bone morphogenetic protein 4 induces the differentiation of brain cancer stem cells, preventing tumor recurrence, and their intranasal or intravenous administration has been shown to improve survival rates in F98 rats [98]. Furthermore, ADSCs can be engineered to express suicide genes such as HSV-tk. Malik et al. utilized PLL-PEI in conjunction with TRAIL and suicide gene strategies to extend the survival in C6 glioma rats [99]. Integrating NSCs with nanoparticles in brain GBM models hastens tumor targeting and reduces nanoparticle clearance [100]. MSCs can cross the BBB of Wistar rats under the action of chemokines and migrate to the tumor region, and become carriers for the delivery of GBM therapeutic drugs. The researchers also observed that MSCs may promote tumor growth by releasing exosomes [101]. However, advances in cell membrane bionic carrier technology can address the shortcomings of natural cells as carriers. MSC and NSC membranes are employed as drug delivery vehicles, retaining protocell tumor-homing and BBB-crossing capabilities while enhancing their crossing and tumor targeting through specific modifications [79].

### 2.5. Passive Diffusion

Small, fat-soluble molecules can form transient pores in the phospholipid bilayer, allowing passive diffusion across the BBB [102], a process influenced by molecular properties that typically restrict the penetration of the majority of molecules, with only a small subset of drugs effectively treating CNS diseases [103]. These hydrophobic molecules exploit the enhanced permeability and retention (EPR) effect for intercellular delivery and passive accumulation in cancer tissues, with passive administration across the BBB occurring via paracellular and transcellular routes tailored to drugs with specific physicochemical properties [104]. Strategies to modulate BBB permeability are essential for passive endocytosis, as demonstrated by the efficacy of fluoroethyl-modified tyrosine kinase inhibitors [105] and the ARTPC nanoplatform loaded with TMZ and surface-functionalized with ApoE (ApoE-ARTPC@TMZ), which utilizes LDLR-mediated transcytosis for enhanced BBB permeation and anti-GBM activity in vivo [106]. In vivo studies have shown that targeted liposomes demonstrate an enhanced circulation time, superior BBB permeation, and GBM accumulation, leading to significant anti-GBM effects and prolonged survival in an intracranial U251-TR mouse model [106]. Cationic liposomes selectively target the tumor vasculature due to their affinity for negatively charged BBB endothelial cells [107]. Conversely, non-targeted lipophilic systems may fail to penetrate the BBB due to its tight junctions. The use of drugs (such as mannitol) or FUS can disrupt BBB connections, thereby enhancing drug penetration [104]. The intravenous administration of microbubbles followed by ultrasound exposure induces localized BBB disruption, facilitating transient drug access to the brain parenchyma [108]. Furthermore, microbubble-mediated FUS improves the targeting of drugs to brain tumors [109], with enhanced efficacy when combined with other delivery methods [108].

Table 1 summarizes the preclinical research findings on the therapeutic potential of NDDSs in the treatment of GBM.

## 3. Alternative NDDS Strategies to Bypass the BBB

While crossing the BBB is essential, there are alternative strategies to bypass it entirely, offering potential solutions for drug delivery to the brain.

### 3.1. Intranasal Administration

By administering drugs via the nasal route, it is possible to circumvent the BBB, evade the first-pass metabolism, and achieve the rapid onset of action [110]. ABC and SLC transporters in the nasal cavity facilitate neurotherapeutic efficacy [111]. This approach highlights the intranasal delivery of diverse drugs and biopharmaceuticals [112]. Nanoparticles target brain tumors via olfactory and trigeminal nerves, crossing the olfactory epithelium [113]. In vivo studies show a reduced peripheral distribution and prolonged survival in brain tumor rats [114]. Chitosan–manganese/gold nanoparticle hybrids enhance the RNA delivery to brain regions [115]. However, the limited targeted volume and specificity of nasal administration can cause toxicity. Combining routes, such as microbubble-mediated FUS with nasal delivery, significantly increases targeted drug delivery in tumors [108], promising a more efficient drug delivery to the brain.

### 3.2. Convection-Enhanced Delivery

Convection-enhanced delivery (CED) is a local administration method used during neurosurgery [116] that involves placing a catheter at the tumor site and using an external pump to establish a pressure gradient for direct drug delivery to the target tissue [117]. The infusion dose does not depend on the molecular size or weight, improving the drug spatial distribution and reducing the systemic toxicity, potentially enhancing patient survival rates at 24 and 36 months [118] and the median survival time [119]. Phase I trials for recurrent high-grade gliomas have confirmed the feasibility and safety of the intracerebral CED of carboplatin [120]. However, in a multicenter Phase III study involving 276 patients with recurrent GBM, there was no significant difference in the median survival when using CED with Cintredekin besudotox (IL13-PE38QQR) and Gliadel wafers (a carmustine implant) [121]. The rich tumor vasculature, interstitial fluid pressure within tumors, limitations in catheter technology, and imaging for drug delivery hinder the reliability and reproducibility of this technique. Drug excretion/absorption and catheter placement issues affect tumor exposure [104]. With technological advancements, CED could become a glioma drug delivery technique, but further trials are needed to confirm its efficacy.

### 3.3. Intracavitary/Intrathecal Drug Administration

Intrathecal drug administration, bypassing the BBB to access the ventricular system, elevates brain drug concentrations but risks tissue damage [122,123]. Compared to intraventricular injection, it is less harmful, and it is being extensively studied for the delivery of large molecules in stroke and neurodegenerative models [124] and is a promising avenue for future antitumor nanomedicine delivery.

## 4. Progress in Clinical Trials

Preclinical research is a pivotal step in the drug development process, providing a solid data foundation for clinical trials and drug marketing by comprehensively assessing the safety, efficacy, and pharmacokinetic properties of drugs in animal models and in vitro systems. This phase of research not only aids in optimizing drug candidates and reducing the risks associated with clinical trials but also ensures that new drugs meet stringent safety and efficacy standards before marketing and comply with regulatory requirements. This section will focus on the progress of clinical trials of nanomedicine drug delivery systems in GBM treatment, systematically analyzing key data on drug safety, PK, and pharmacodynamics, and discussing the advantages and disadvantages of different drug delivery systems, as well as the challenges and solutions encountered in clinical trials.

Building upon the robust foundation laid by preclinical research, the field of nanomedicine has made inroads into clinical applications, as evidenced by the favorable outcomes of Phase I clinical trials. Table 2 offers a comprehensive overview of pivotal trials that have utilized NDDSs based on liposomal formulations, polymeric nanoparticles, and inorganic nanoparticles, which have been the subject of considerable interest and investigation [125]. Notably, the clinical progression of nanoliposomal irinotecan, assessed in trials NCT00734682, NCT02022644, NCT03086616, and NCT03119064 since 2008, underscores the extensive exploration of nanomedicine in GBM treatment. A notable Phase I trial among recurrent GBM patients is aimed at ascertaining the maximum tolerated dose and preliminary therapeutic effects of PEGylated nanoliposomal irinotecan in conjunction with TMZ. Despite the study’s premature conclusion due to unmet efficacy expectations in an interim analysis [126], it has yielded critical insights for the development of advanced treatment modalities. Another Phase I trial investigated the safety, PK, and maximum tolerated dose of intravenously administered nal-IRI, considering the UGT1A1*28 genotype. The findings suggest that intravenous nal-IRI is well tolerated, and that the genotype does not significantly alter the drug’s PK or maximum tolerated dose [127]. The next step involves exploring the efficacy and safety of nal-IRI under CED, though the outcomes of this trial have yet to be published (NCT02022644). The results of these clinical trials are anticipated to provide crucial information for the clinical translation of nanomedicine therapies for GBM. Furthermore, Phase II clinical trials are ongoing to investigate the efficacy and safety of other liposomal NDDSs. For instance, the combination of PEGylated liposomal doxorubicin (PLD) and TMZ has demonstrated superior outcomes compared to radiotherapy alone in preliminary studies [128]. Although the sample size was limited and lacked randomized controls, this finding warrants further investigation.

To enhance the NDDSs’ targeting efficiency, chemical modification has emerged as a strategy aimed at precisely targeting GBM cells, thereby bolstering the safety and efficacy of treatment. For instance, Kasenda et al. reported the effective drug delivery capability of anti-EGFR doxorubicin-loaded immunoliposomes in tumor tissues with a compromised BBB [131]. EGFR-Erbitux receptor EnGeneIC Dream Vector with mitoxantrone (EEDVsMit), an innovative nanocellular therapy, has demonstrated good tolerability and no dose-limiting toxicity in clinical trials [133]. This advancement offers novel therapeutic strategies for pediatric refractory tumors, particularly those with positive EGFR expression. Additionally, glutathione, as a targeting ligand, has been conjugated to polyethylene glycol liposomes, a strategy designed to enhance the drug delivery efficiency to the brain. Phase I/II clinical trials on glutathione PEGylated liposomal doxorubicin (2B3-101) were completed in 2014 (NCT01386580).

In the research on strategies to enhance NDDS targeting, the utilization of biological vectors, particularly human cells, has become a key innovation for achieving the precise localization of GBM cells and enhancing the safety and efficiency of treatment. Genetically engineered NSCs that express cytosine deaminase (CD) can efficiently convert 5-fluorocytosine (5-FC) into 5-fluorouracil (5-FU) in the tumor microenvironment. Portnow et al.’s study first demonstrated the safety of a single intracranial injection of CD-NSCs after the oral administration of 5-FC in recurrent high-grade GBM patients [134]. Subsequent studies further evaluated the safety of the repeated intracranial injection of CD-NSCs and explored the appropriate dose when combined with leucovorin, laying the foundation for Phase II clinical trials [135]. This strategy significantly enhances the targeting of treatment by bypassing the BBB, optimizing the drug concentration in the affected area, and reducing systemic side effects. NSCs, as ideal biological vectors, have the potential to deliver a variety of anticancer therapies to the tumor region. For example, in clinical trials using NSCs to deliver oncolytic adenovirus (NSC-CRAd-S-pk7) to newly diagnosed malignant GBM patients, a good safety was observed in the dose-escalation phase, and a partial tumor volume reduction and other positive responses were observed in some patients [136]. Another clinical trial is using hCE1m6-NSCs expressing carboxylesterase and irinotecan to treat recurrent high-grade GBM to inhibit tumor growth (NCT02192359), aiming to determine the recommended Phase II dose, with the results yet to be released. Additionally, monocyte antigen carrier cells have shown potential in the treatment of newly diagnosed GBM with an unmethylated MGMT gene promoter, such as the MT-201-GBM vaccine, which is made from a patient’s own cells loaded with CMV pp65-LAMP mRNA and which has been in clinical trials since August 2023 to determine its maximum tolerated dose (NCT04741984). Meanwhile, antigen MRNA-loaded dendritic cells (DCs) have also entered clinical trials using autologous DCs as mRNA delivery carriers to target survivin for GBM of the brain (NCT06524063). This trial is currently in progress, and the results have not yet been published. Combining the dual strategy of using genetic engineering technology and biological vectors is expected to further promote the development of the field of GBM treatment and bring hope to more patients.

Lastly, inorganic nanoparticles composed of magnetically responsive materials, such as superparamagnetic iron oxide nanoparticles (NanoTherm^®^), have shown potential in prolonging the overall survival in patients with recurrent GBM [129]. In 2013, NanoTherm^®^ became the first nanotechnology-based cancer treatment device to receive CE mark approval. Furthermore, a new clinical trial (NCT06271421) is evaluating the efficacy and tolerability of the NanoTherm treatment system in recurrent GBM via cyclic hyperthermia. Meanwhile, another type of inorganic nanoparticle, NU-0129 (NCT03020017), a nanomedicine delivery system centered on gold nanoparticles and covalently conjugated with small interfering RNA (siRNA) oligonucleotides, targets the BCL2L12 gene in GBM cells. This system is being explored through clinical trials for its potential as a brain-penetrating precision therapy [130]. Additionally, a Phase I/II clinical trial investigating AGuIX nanoparticles combined with radiotherapy and temozolomide in newly diagnosed GBM patients has shown that AGuIX nanoparticles significantly enhance the clinical potential of GBM treatment when used in conjunction with radiotherapy and temozolomide [132].

Since the clinical application of nanomedicine delivery systems over a decade ago, numerous Phase I/II clinical trials have been conducted to comprehensively evaluate their safety and efficacy, yet the outcomes of most of these trials remain unpublished. Among the published results, the strategy of encapsulating drugs within liposomes and delivering them via passive diffusion mechanisms has demonstrated a more frequent application compared to other administration routes. Despite summarizing eight strategies to overcome the BBB from preclinical studies, this preference is understandable in that it may be attributed to the generally good liposolubility of first-line drugs commonly used in GBM treatment, coupled with the unique advantages of liposomes as carriers, including a high stability, low leakage rates, sustained drug release, and excellent biocompatibility. Building upon this foundation, scientists have further introduced ligands into liposomal formulations to enhance drug targeting. Based on the outcomes of two reported clinical trials, targeted immunoliposomes have shown potential in delivering cytotoxic drugs and potentially immunomodulatory molecules to GBM tissues, albeit without achieving the anticipated significant clinical improvement. However, the exceptional long-term remissions observed in individual patients provide positive signals for this novel therapeutic strategy, hinting at its potential positive impact on treatment outcomes [131]. The utilization of cells or cell membranes as drug delivery vehicles constitutes a prominent option in clinical translation, with notable achievements particularly evident in studies employing neural stem cells (NSCs) as carriers, which have yielded published clinical outcomes. The dual-pronged strategy of combining gene engineering technologies with NSCs as biological vectors has yielded more abundant clinical translation results compared to other cell types. This may stem from the unique advantages possessed by NSCs that are not shared by other cell types. For instance, NSCs exhibit a superior BBB permeability, precise migration and tracking capabilities towards tumor regions, and a differentiation potential that aids in brain tissue repair and regeneration, alongside relatively low immunogenicity. In contrast, the use of red blood cells as drug carriers is hindered by limitations such as burst release characteristics and the absence of specific receptors for targeted delivery.

Given the challenges faced in GBM chemotherapy, the ongoing proliferation of clinical trials signifies the gradual progress in the clinical translation of nanomedicines. In the future, the key to nanomedicine development will lie in optimizing drug delivery systems, enhancing drug targeting and efficacy, and actively exploring combination strategies with other treatment modalities (such as immunotherapy and radiotherapy) to achieve more pronounced therapeutic effects.

## 5. Technical Challenges and New Strategies

The BBB comprises endothelial cells, pericytes, astrocytes, the basement membrane, tight junctions, and adherens junctions among endothelial cells [103,137] (Figure 2a). The intact BBB safeguards the brain from toxins while restricting drug entry. Only small, lipophilic molecules (<500 daltons) can cross it [138]. The coordinated action of three cell types constructs a robust “city wall” for brain protection, utilizing membrane receptors, ion channels, and transport proteins as “delivery channels”. To deliver drugs, leveraging these channels is key. However, the efflux pumps on cell membranes pose a challenge. In gliomas, the BBB integrity deteriorates with tumor progression, influenced by the malignancy grade [72]. An elevated malignancy correlates with heightened metabolic demands, yet nutrient and oxygen scarcity induce local hypoxia, fostering abnormal angiogenesis and BBB dysfunction [10]. Despite this variable BBB integrity, local disruptions do not markedly affect drug concentrations in tumor tissues [139,140]. Consequently, BBB penetration is essential for effective anti-glioma drug design.

The main barriers for therapeutic drugs to penetrate the BBB encompass four aspects: the “physical barrier”, “transportation barrier”, “metabolic barrier”, and “immune barrier”.
(1)The “physical barrier”: This barrier encompasses the anatomical and functional features of BBB endothelial cells, forming a vital anatomical gateway for targeted brain drug delivery. The lipid bilayer membranes of these cells exhibit lipophilicity and host receptors, carrier proteins, and other components that regulate molecular trafficking from the bloodstream to the brain tissue. High-molecular-weight drugs (>500 daltons) often fail to traverse this barrier [139]. The tight and adherens junctions between endothelial cells maintain the BBB’s integrity, preventing unrestricted substance exchange;(2)The “transportation barrier”: Functionally, the surfaces of BBB endothelial cells are negatively charged, impeding negatively charged compounds from entering neurons. Endothelial membranes express specific transporters that regulate substrate influx/efflux, preventing unauthorized bloodstream substances from crossing [141]. Pericytes and astrocytes encapsulate BBB endothelial cells, creating resistance that allows only small molecules (e.g., water, gases, lipids) to diffuse passively. Large, charged, polar, hydrophilic molecules (amino acids, glucose, drugs) rely on luminal membrane transport proteins/receptors [137]. ATP-driven efflux pumps (P-glycoprotein) limit toxin/drug permeability, reducing CNS exposure [103,142], impacting drug efficacy, exacerbating side effects, and challenging the drug action in brain tissue;(3)The “metabolic barrier”: Various drug-metabolizing enzymes, such as CYP450 enzymes, have been documented within the endothelial cells of the brain [103];(4)The “immune barrier”: Neurovascular units, comprising pericytes and astrocytes, regulate tight junctions, waste clearance, the vascular function, and neuroimmune responses, forming an “immune barrier” that constitutes the BBB.

The main challenges in BBB drug delivery involve physical, transportation, metabolic, and immune barriers. The rational design of drug delivery systems, particularly nanocarriers, can overcome metabolic and immune barriers by reducing enzymatic reactions, phagocytosis, and immune clearance, enhancing drug stability [143]. From the perspective of BBB morphology and function, there are two types of coping strategies to overcome the BBB: crossing it and bypassing it. This review emphasizes how nanodrug delivery systems penetrate the BBB, focusing on overcoming physical and transportation obstacles.

## 6. Conclusions and Perspectives

The BBB plays a pivotal role in influencing the efficacy of GBM treatment agents. NDDSs have ushered in a new era for brain drug delivery, potentially overcoming the BBB to improve the treatment outcomes of GBM patients. Preclinical studies reveal that NDDSs exhibit diverse delivery modalities and superior delivery efficiencies through their rational design compared to drug substances. Encapsulation and specific chemical modifications enhance drug stability and targeting, thereby significantly mitigating toxicity and side effects, advancing towards clinical translation. Nevertheless, the number of NDDSs transitioning into clinical practice remains limited. In contrast, emerging therapies such as tumor immunotherapy and cellular therapy are progressively being explored, posing additional challenges to the widespread clinical application of NDDSs. Consequently, continued technological innovation aimed at markedly improving the clinical efficacy of NDDSs is imperative for achieving their broad clinical transformation. Another challenge that lies ahead is the complexity and heterogeneity of GBM, which further require that researchers integrate knowledge from multiple disciplines and undergo a paradigm shift in the strategic development of NDDSs, to drive them from the laboratory to the clinic [144].

In clinical trials, in order to improve the clinical efficacy of NDDSs, researchers tend to adopt the combination therapy strategy [132], aiming to achieve a synergistic effect in three primary avenues, thereby attaining a “1 + 1 > 2” augmentation: (1) Theranostic NDDSs. Endowing nanomedicines with the dual functions of therapy and imaging provides a robust tool for early tumor diagnosis, treatment monitoring, and efficacy assessment. A feasible approach is the co-assembly of magnetic resonance imaging contrast agents and therapeutic drugs within NDDSs, enabling real-time navigation for multimodal theranostics [145]. (2) The combination of therapeutic modalities, for instance, utilizing CAR T cells and CAR neutrophils for targeted drug delivery. The integration of chemotherapy and immunotherapy mitigates off-target effects and prolongs the survival of tumor-bearing mice [85,86,90]. (3) The design of multi-drug synergistic NDDSs. The combined delivery of sicPLA2 and metformin based on exosomes selectively targets the GBM energy metabolism for antitumor effects, demonstrating the potential for personalized treatment in GBM patients [146]. However, these strategies are currently in preclinical research. Despite the aforementioned challenges, recent clinical GBM trials using nanomedicine as a therapeutic modality indicate the gradual expansion of the clinical application of nanochemotherapy in GBM treatment.

Looking ahead, further advancements in NDDSs will require interdisciplinary collaboration across fields such as nanotechnology, neurobiology, and clinical medicine. Future research should focus on optimizing nanoparticle design for specific therapeutic applications, improving patient-specific targeting, and refining drug release mechanisms. Additionally, well-designed clinical trials are crucial for evaluating the safety, efficacy, and long-term outcomes of these innovative delivery systems. By addressing these challenges, NDDSs can move closer to realizing their full potential in GBM therapy, ultimately leading to more effective treatments with fewer side effects and better patient outcomes.

## Figures and Tables

**Figure 1 cancers-16-03300-f001:**
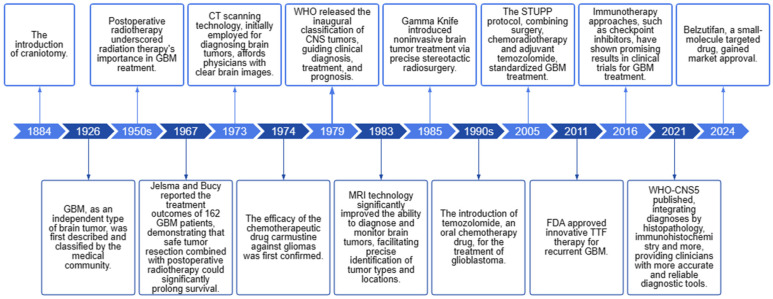
Timeline of important events in the history of GBM research and treatment.

**Figure 2 cancers-16-03300-f002:**
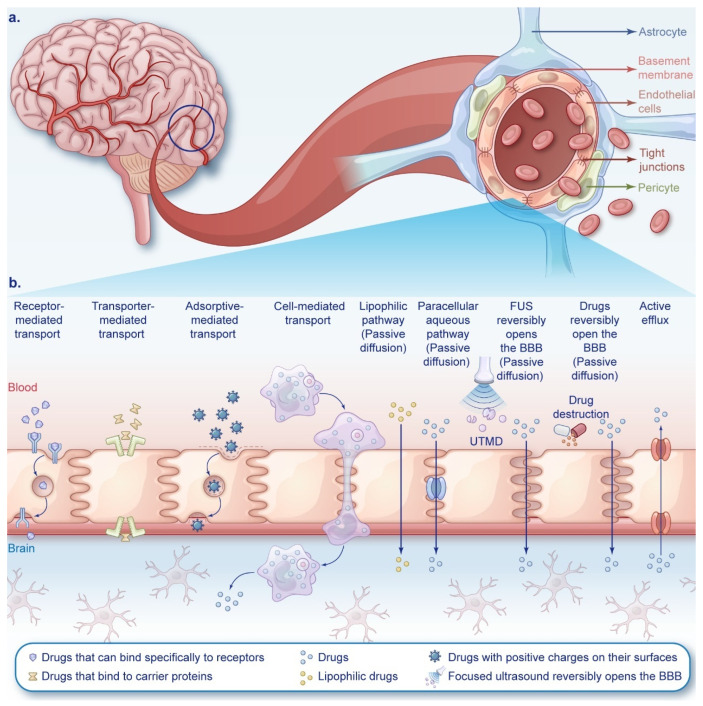
Schematic diagram of the BBB structure and the mechanism of drug entry and exit from the BBB. (**a**) Schematic cross-section of the BBB. (**b**) Schematic diagram of different mechanisms for BBB crossing. From left to right is receptor-mediated transport; transporter-mediated transport; adsorptive-mediated transport; cell-mediated transport; the lipophilic pathway (passive diffusion); the paracellular aqueous pathway (passive diffusion); focused ultrasound (FUS) reversibly opens the BBB (passive diffusion); drugs reversibly open the BBB (passive diffusion); active efflux.

**Figure 3 cancers-16-03300-f003:**
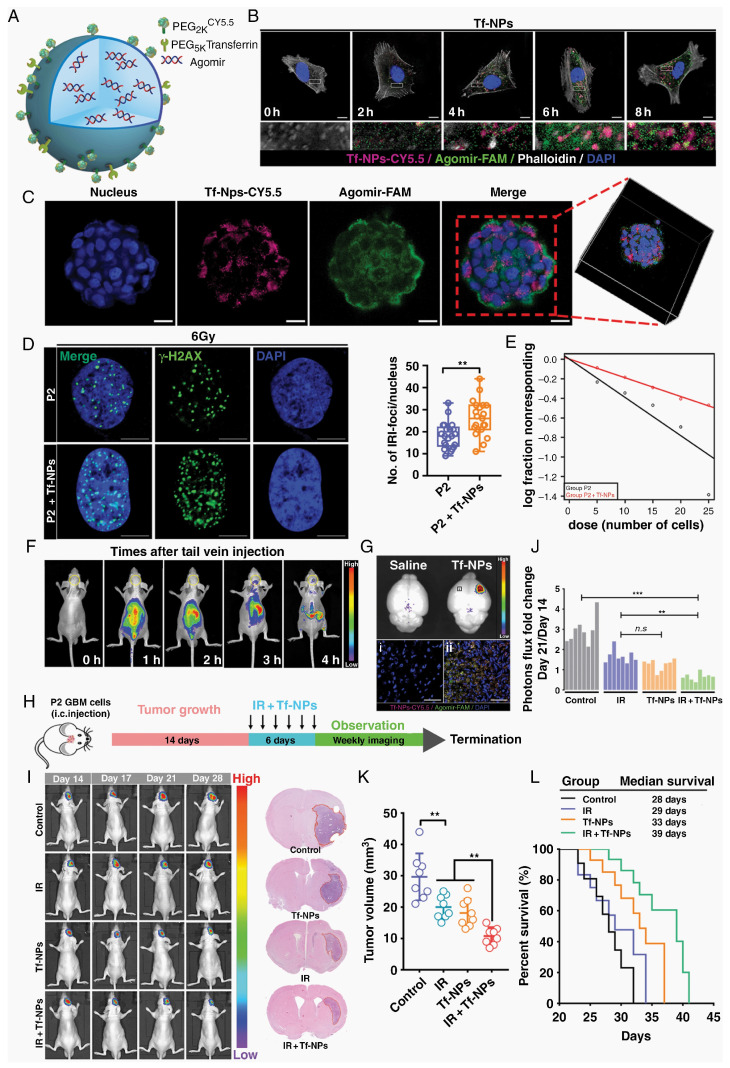
Receptor-mediated transport. RTP-targeting Tf-modified nanoparticles reverse refractory GBM and improve radiosensitivity. (**A**) Schematic of PEGylated agomir-loaded nanoparticles that can be functionalized to enhance transport across the BBB and target RTP cells. (**B**) Immunofluorescence staining demonstrating the time-dependent intracellular uptake of Tf-NPs in P2 cells. Scale bar: 10 μm. (**C**) Tf-NPs entering a GSC spheroid were monitored via 3D confocal laser microscopy. Scale bar: 20 μm. (**D**) P2 cells were treated with IR in the presence or absence of Tf-NPs, and γ-H2AX focal formation was investigated. Scale bar = 10 μm. (**E**) In vitro limiting dilution assay. (**F**,**G**) In vivo real-time NIR fluorescence imaging of P2 tumor-bearing mice after Tf-NP administration for indicated time periods. (**H**) Schematic diagram showing experimental time course and details of Tf-NP and IR treatment courses. (**I**) In vivo bioluminescence images of P2 tumor cells in orthotopic mice intravenously injected with Tf-NPs. (**J**) Statistical analysis of orthotopic tumor growth from P2 cells. (**K**) Quantification of tumor sizes. The data were obtained from H&E-stained brain sections of 8 mice per group. (**L**) Kaplan–Meier survival curves of mice intracranially injected with P2 cells (** *p* < 0.01, *** *p* < 0.001, *n.s p* > 0.05) [12]. Copyright 2022, Oxford University Press.

**Figure 4 cancers-16-03300-f004:**
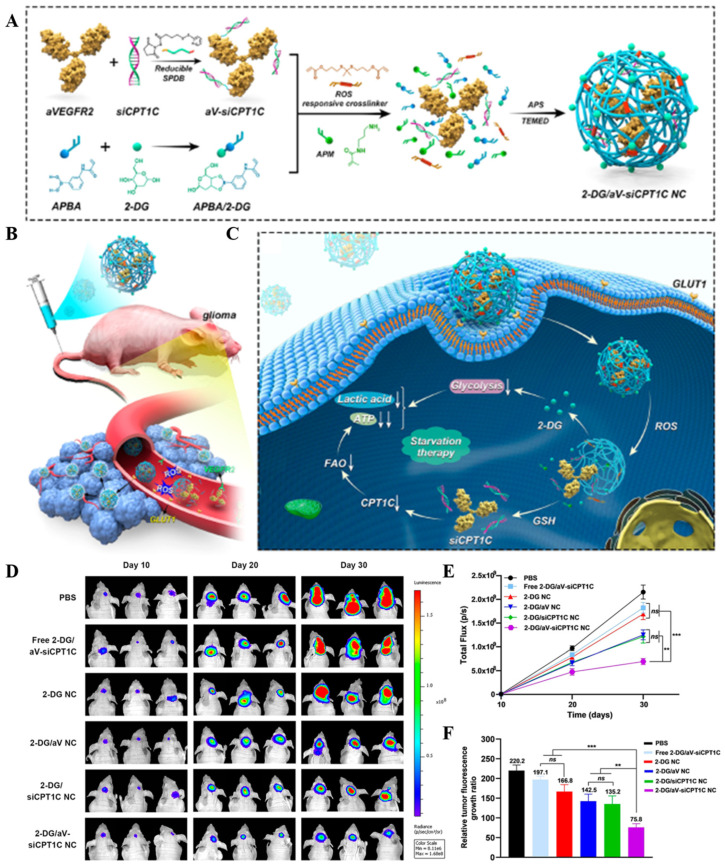
Transporter-mediated transport. (**A**–**C**) The 2-DG/aV-siCPT1C NC traverses the BBB via GLUT1 receptors to enter the CNS, thereby achieving starvation therapy for the GBM. (**D**–**F**) Biofluorescence imaging of U87-Luci glioma mice with different treatments, the quantification of glioma bioluminescence signals in each group, and the relative tumor fluorescence growth ratio for each treatment (*n* = 3, ** *p* < 0.01, *** *p* < 0.001, *ns p* > 0.05) [39]. Copyright 2023, American Chemical Society.

**Figure 5 cancers-16-03300-f005:**
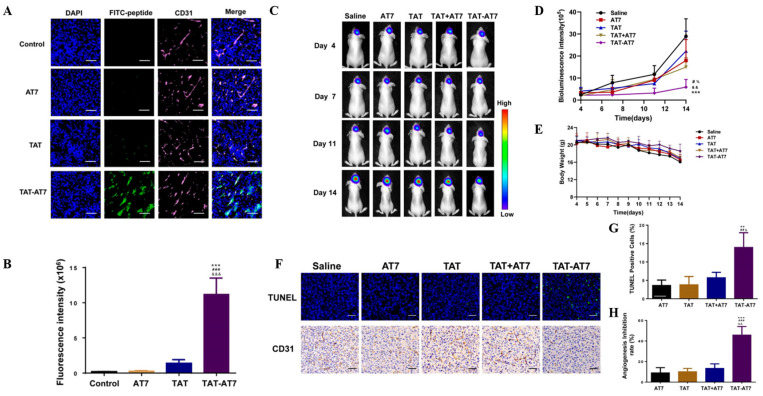
Adsorptive-mediated transport. (**A**–**B**) TAT-AT7 targeted glioma blood vessels in the intracranial glioma tissue of nude mice. Bars represent 200 μm. (**C**–**H**) TAT-AT7 inhibited the growth of gliomas in nude mice. Bars represent 100 μm (** *p* < 0.01, *** *p* < 0.001 vs. AT7 group; # *p* < 0.01, ## *p* < 0.01, ### *p* < 0.001 vs. TAT group; & *p* < 0.05, && *p* < 0.01, &&& *p* < 0.001 vs. TAT + AT7 group, % *p* < 0.05 vs. TAT + AT7 group) [65]. Copyright 2023, MDPI.

**Figure 6 cancers-16-03300-f006:**
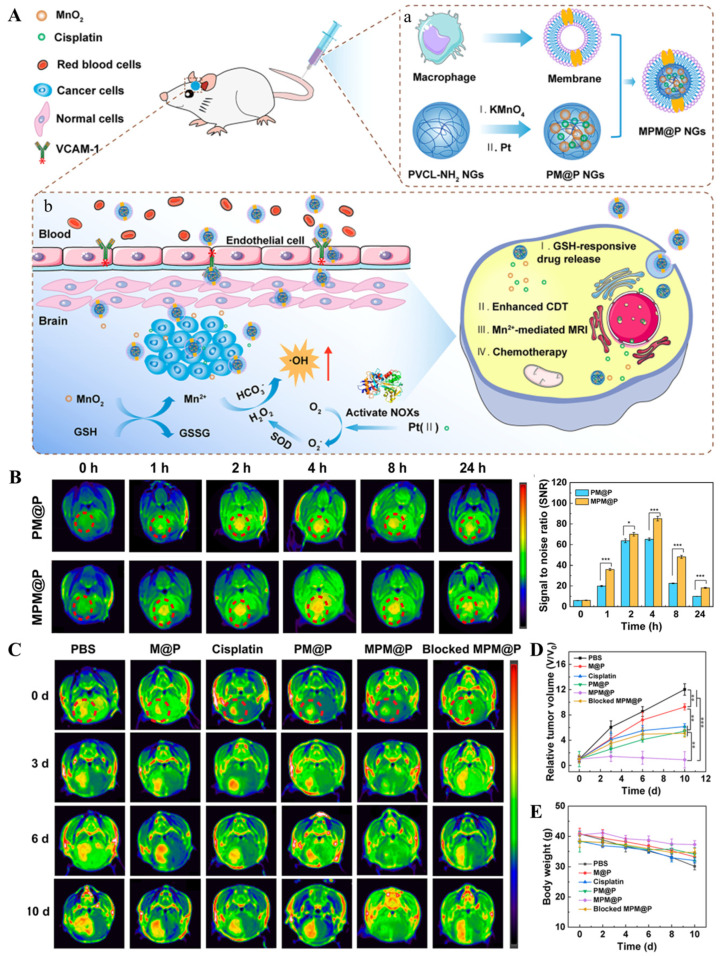
Cell-mediated transport. (**A**) Preparation of macrophage membrane-based biomimetic vehicles and mechanism of biomimetic MPM@P NGs crossing BBB to penetrate brain parenchyma for MR imaging-guided combination chemotherapy/chemodynamic orthotopic glioma therapy. (**B**) The targeted MR imaging of orthotopic glioma verified that macrophage membrane decoration facilitated the traversal of MPM@P NGs across the BBB. (**C**–**E**) Changes in tumor volume and body weight in C6 glioma-bearing mice after different treatments (* *p* < 0.05, ** *p* < 0.01, *** *p* < 0.001) [75]. Copyright 2021, American Chemical Society.

**Table 1 cancers-16-03300-t001:** Preclinical research achievements of NDDSs in GBM treatment.

BBB-Crossing Strategy	Targeting	NDDSs	Drug	Combination Therapy (If Any)	Administration Route	Administration Model	Safety Evaluation	Pharmacokinetics (PK)	Pharmacodynamics	Ref.
RMT	Tf	Tf-NPs	TMZ	Bromodomain inhibitor	Tail vein injection	Human U87MG and murine GL261 glioma models	Mice treated with drug-loaded Tf-NPs maintained their body weights and body conditioning scores and had significantly more stable WBC and PLT levels over the treatment course compared with the control group.	The 24 h absorption efficiency was higher (13% vs. >1%). Tf-NPs accumulated and were retained on the tumor surface, while non-functional NPs were not. NPs loaded with TMZ demonstrated an attenuated release profile with ~90% release by 48 h.	Tf-NPs bind to GBM tumors, enhance DNA damage and apoptosis, reduce tumor burden, improve survival, and protect against systemic toxicity.	[13]
RMT	Lf	PMO-Lf@Dox	DOX	-	Co-incubation	C6 cells	PMO concentration was 200 μg/mL, cell survival rate was 90%, and hemolysis rate was <2%. One week after tail vein injection of 60 mg/kg PMO, no obvious changes were seen in heart, liver, spleen, lungs, or kidneys.	PMO-Lf@Dox-treated C6 cells showed a significantly higher uptake after 24 h than PMO@Dox-treated cells.	Lf-modified PMO enhances the inhibitory effect of Dox on C6 cells.	[15]
RMT	Lf	USLP-NH_2_-PEG-TMZ-Lf	TMZ	-	Tail vein injection	Transwell model with hCMEC/D3 cells; Balb/C mice	Safety was evaluated via H&E staining of mouse organs (brain, kidneys, lungs, liver) post-nano-formulation IV injection, with no toxicity observed in 24 h.	USLP-NH2-Cy5-PEG-Lf accumulated rapidly in the brain and reached its peak at 1 h post-administration, and the USLP formula significantly reduced the external alignment.	In vitro apoptosis studies on GBM cell lines U87 and GL261 showed improved TMZ-induced apoptosis with USLP formulations compared to pure TMZ.	[16]
RMT	Acetylcholine receptor	TMZ@RVG-Zein NPs	TMZ	-	Co-incubation	U87 cell lines; BBB model with bEnd.3 cells	TMZ@RVG-Zein NPs exhibited excellent stability, without causing significant side effects.	The TMZ@RVG-Zein NPs had an encapsulation efficiency (EE) of 77.9 ± 4.7% and a loading efficiency (LE) of 66.7 ± 2.9 mg/g.	TMZ@RVG-Zein NPs had cytotoxic effects on U87 cells and induced apoptosis and showed enhanced cellular uptake compared to TMZ alone.	[17]
RMT	FR	iRPPFFA@TMZ	TMZ	-	Tail vein injection	Subcutaneous and orthotopic xenograft tumor models	The biocompatibility of POSS nanoparticles was evaluated via a CCK-8 assay, hemolysis rate tests, and fluorescence imaging, showing a low toxicity and effective cellular uptake for biomedical applications.	The multifunctional POSS nanoparticles demonstrated high stability with no significant fluorescence intensity change over 8 weeks, while the iRPPFFA nanoparticle half-life was estimated at approximately 16 weeks.	In vivo studies showed that TMZ-loaded POSS nanoparticles significantly improved the survival of GBM-bearing mice, indicating an enhanced therapeutic efficacy compared to monotherapy.	[19]
RMT	LRP-1 receptors	Ti@FeAu–Ang nanoparticles	Magnetic nanoparticles as therapeutic agents	-	Tail vein injection	Rat GBM model	Preliminary safety analysis highlighted no toxicity to the hematological system after Ti@FeAu–Ang nanoparticle-induced hyperthermia treatment. Immunohistochemical analysis showed no significant organ damage or biological changes in vital organs such as the heart, liver, spleen, lungs, and kidneys.	The paper does not provide detailed pharmacokinetic data, but it does mention that the nanoparticles were prepared at various concentrations for the temperature elevation test and in vivo tumor growth assessment.	The Ti@FeAu–Ang nanoparticles demonstrated a temperature elevation of up to 12 °C upon magnetic stimulation, indicating potential applications in MRI- and hyperthermia-based cancer therapy. The nanoparticles showed improved cytotoxicity up to 85% in vitro due to hyperthermia produced by a magnetic field. In vivo findings showed a 10-fold decrement in tumor volume compared to that of the control group.	[22]
RMT	LRP-1 receptors	Au-DOX@PO-ANG	DOX	Radiotherapy	Tail vein injection	U87-MG human GBM xenografts in nude mice	Blood biochemical indicators (CK-MB, AST, and Scr) were measured in vivo, and no significant pathological changes were observed in the main organs (heart, liver, spleen, lungs, or kidneys) of the Au-DOX@PO-ANG group compared to the PBS group. The cytotoxicity of the modified AuNPs was evaluated in vitro using CCK-8 assays, showing more than 80% cell viability at concentrations up to 10 mg/L, indicating non-toxicity to GBM cells.	The storage stability of the modified AuNPs in PBS (pH 7.4) at 4 °C showed little particle size change over 4 weeks, indicating good storage ability and potential stability.	The antitumor activity of Au-DOX@PO-ANG was evaluated in vitro using the CCK-8 assay, showing significant antitumor effects when combined with radiotherapy. The therapeutic effect was observed in vivo through MRI, with the tumor volume rate of increase slowing in the Au-DOX@PO-ANG + RT group, and a significant increase in cell apoptosis was observed in this group, consistent with the MRI data.	[23]
RMT	EGFR	PmAb-TMZ-PLGA-NPs	TMZ	Panitumumab	Co-incubation	U-87MG and LN229 GBM cell lines	In vitro cytotoxicity assessed using the Live/Dead assay kit and fluorescence-activated cell sorting (FACS). Immunoreactivity evaluated using EGFR-overexpressed U-87MG cells.	In vitro drug release was studied using phosphate-buffered saline (PBS) under acidic (pH 5.0) and neutral (pH 7.4) conditions. Release profile determined via UV–Vis spectroscopy at various time points.	In vitro cytotoxicity/drug release was enhanced in U-87MG cells due to high EGFR expression compared to LN229 cells.	[25]
RMT	IGF1R	IGF1R4-mFc	Galanin peptide	-	ISBP	Rat model; Hargreaves pain model	No specific safety evaluation was described in detail. However, the use of sdAbs, which are known for their low immunogenicity and high solubility/stability, suggests potential safety advantages.	IGF1R4-mFc showed significant brain uptake compared to the negative control A20.1-mFc, with ~25% of the total amount accumulating in the brain parenchymal fraction post-ISBP. The concentration curve for IGF1R4-mFc demonstrated a linear accumulation plateauing at approximately 400 µg (~1 µM), suggesting a saturable transport mechanism.	The systemic administration of IGF1R4-mFc fused with galanin induced a dose-dependent suppression of thermal hyperalgesia in the Hargreaves pain model, indicating the pharmacological effectiveness of the brain-delivered cargo.	[26]
RMT	α5β1 integrin receptor	RGEK-lipopeptide containing coencapsulated STAT3siRNA and WP1066	WP1066; STAT3siRNA		Tail vein injection	Intracranial orthotopic GL261 GBM model in C57BL/6J male mice	The document does not provide direct safety evaluation data. However, the use of integrin receptor-selective liposomes and their preferential accumulation in tumor tissue suggests the potential for reduced systemic toxicity.	NIR dye-labeled α5β1 integrin receptor-selective liposomes were found to accumulate preferentially in mouse brain tumor tissue after intravenous administration. Encapsulation efficiency: entrapment efficiency (%EE) for WP1066 was measured using analytical HPLC.	The coadministration of WP1066 and STAT3siRNA within RGDK-lipopeptide-based liposomes led to significant inhibition (>350% compared to the untreated mouse group) of orthotopically growing mouse GBM.	[30]
RMT	αv β integrin and NRP-1	CPT-S-S-PEG-iRGD@IR780 micelles	Camptothecin (CPT)	Photodynamic therapy	Tail vein injection	Intracranial orthotopic U87MG glioma tumor model in Balb/c nude mice	The micelles were shown to be stable with controlled drug release under physiological conditions. Toxicity studies in vivo assessed the mice’s survival, body weights, and histopathologies.	Information on the pharmacokinetic behavior of the micelles is not explicitly detailed in the Abstract. However, the micelle design enabled sustained CPT release upon exposure to high glutathione levels in glioma cells.	The CPT-S-S-PEG-iRGD@IR780 micelles displayed significantly enhanced antitumor effects with laser irradiation, as compared to controls. Micelles with iRGD demonstrated favorable targeting ability to glioma cells and deep tumor penetration.	[34]
RMT	Hsp70	D-A-DA/TPP	DOX	PD-1 checkpoint blockade	Tail vein injection	C6-luc tumor-bearing mice	In vivo toxicity assessment: safety evaluation likely conducted by monitoring animal health, blood chemistry, and tissue histology post-treatment. However, specific details are not provided.	The biodistribution and clearance of D-A-DA/TPP nanoparticles in vivo are not explicitly discussed in the excerpt. These would typically involve measuring nanoparticle concentrations in blood, tumors, and other organs over time.	The efficacy of D-A-DA/TPP nanoparticles at inducing glioma apoptosis and prolonging the median survival time was demonstrated through in vivo studies. Combination with PD-1 checkpoint blockade was shown to further activate T cells and provoke an antitumor immune response.	[35]
RMT	Dopamine and GRP78 receptors	pHA-AOHX-VAP-DOX	DOX	-	Tail vein injection	Intracranial U87 glioma-bearing nude mice model	The maximum tolerated dose (MTD) was determined in healthy BALB/c mice. Hematoxylin–eosin (H&E) stains of major organs and blood samples were collected to measure blood routine and biochemical parameters.	The conjugate biodistribution in tumor and normal tissues was evaluated. The DOX accumulation at the tumor site was assessed, and the conjugation with peptides reduced the DOX accumulation in normal tissues.	The anti-GBM efficacy was evaluated by the prolonged survival time of mice and assessing the tumor cell apoptosis through TUNEL staining. The inhibition of tumor angiogenesis was evaluated using CD31 immunofluorescence staining.	[36]
TMT	GLUT1	2-DG/aV-siCPT1C NC	2-DG	siCPT1C	Tail vein injection	Orthotopic U87-Luci cell xenograft tumor model in BALB/c nude mice	The nanocapsule safety was evaluated through histopathologic studies and blood biochemical tests, including ALT, AST, BUN, and CREA levels. No significant toxicity or side effects were observed on normal cells and tissues.	The nanocapsules showed an extended half-life compared to free siRNA, with the 2-DG/aV-Cy5-siCPT1C NC having an elimination half-life (t1/2) of approximately 1.2 h.	The nanocapsules effectively targeted GBM, inhibited energy metabolism, and showed a significant inhibitory effect on GBM growth. The combination of 2-DG, aV, and siCPT1C resulted in decreased lactic acid levels and reduced ATP production in tumor tissues, indicating effective metabolic pathway inhibition.	[39]
TMT	ChTs	pMPC-co-(anti-PD-L1-pPEGMA)	Anti-PD-L1	-	Intravenous injection	LCPN orthotopic glioma tumor model	The safety and biocompatibility of the delivery system were evaluated through in vitro cytotoxicity studies using bEnd.3 and LCPN cells, as well as via a histological examination of major organs from mice.	The PK of the Cy5.5-labeled IgG in the anti-PD-L1-MP-3 formulation was studied, showing a prolonged blood circulation time compared to that of non-choline-containing controls.	The system demonstrated pH-responsive protein release in vitro, with accelerated release at pH 6.0 simulating the acidic tumor microenvironment. In vivo studies showed significant tumor growth suppression and prolonged animal survival, indicating the activation of antitumor immune responses.	[46]
TMT	LAT1	WP1066-loaded liposomes of Amphi-DOPA	wp1066	DNA vaccine	Tail vein injection	Orthotopic GL261 tumor model in female C57BL/6J mice	The safety and toxicity of the liposomal formulation were evaluated through in vivo serum toxicity profiles. No significant changes in the biochemical or hematological parameters suggest that the system is well tolerated.	Intravenously administered NIR dye-labeled Amphi-DOPA liposomes showed a preferential accumulation of the dye in brain tissue, indicating successful BBB penetration.	WP1066-loaded Amphi-DOPA liposomes alone showed an enhanced overall survivability of C57BL/6J mice bearing orthotopically established mouse GBMs by ~60% compared to untreated mice. Combination therapy further enhanced the overall survivability (>300% compared to untreated mice) when combining WP1066-loaded Amphi-DOPA liposomes with in vivo DC-targeted DNA vaccination using a survivin-encoded DNA vaccine.	[50]
TMT	SVCT2	PTX-Glu-Vc-Lip	PTX	-	Tail vein injection	Intracranial C6 glioma-bearing mice	The safety of the ligand-modified liposomes was demonstrated through hemolysis assays, showing no significant increase in hemocompatibility even at high phospholipid concentrations.	The study evaluated the plasma concentration–time profiles and brain distribution of paclitaxel after the intravenous injection of different liposome formulations. The pharmacokinetic parameters, including the AUC(0-t), MRT, Tmax, Cmax, and t1/2, are reported for each formulation.	The cellular uptake of the liposomes was evaluated in GLUT1- and SVCT2-overexpressed C6 cells, showing higher uptake for the Glu-Vc-Lip compared to other formulations. The in vivo imaging of DiD-loaded liposomes demonstrated the targeting efficiency to the brain tumor site.	[53]
TMT	OCTN2	LC-1000-PLGA NPs	PTX	-	Tail vein injection	2D and 3D tumor growth models using glioma cell line T98G	The specific dose is not explicitly mentioned in the provided text. However, the studies involved the use of different paclitaxel concentrations in the in vitro cytotoxicity assays and varying PEG spacer lengths in the nanoparticles.	LC-PLGA NPs showed high accumulation in the brain as indicated by biodistribution and imaging assays in mice. Paclitaxel-loaded LC-PLGA NPs showed the sustained release of paclitaxel compared to Taxol.	The pharmacodynamic evaluation included in vitro cytotoxicity assays in T98G cells, demonstrating increased toxicity with LC-PLGA NPs compared to Taxol- and paclitaxel-loaded PLGA NPs. In vivo biodistribution studies showed an enhanced brain accumulation of paclitaxel with LC-1000-PLGA NPs.	[56]
TMT	Cx43 and BSAT1	Cx43-NG/CDDP and BSAT1-NG/CDDP	Cisplatin	-	Tail vein injection	Intracranial implantation of rat GBM 101/8 in female Wistar rats	Safety was evaluated by monitoring body weight changes and the general condition of the animals, and by comparing the median survival rates of the different treatment groups.	The nanogel PK was assessed by monitoring the tumor volume changes over time using MRI and comparing the median survival times of the treated groups with those of the control group.	The antitumor efficacy of the targeted nanogels was evaluated by comparing the glioma volume and the survival rate of rats treated with targeted nanogels conjugated with specific mAbs against Cx43 and BSAT1 to those treated with non-targeted nanogels or free cisplatin. The study demonstrated a significantly reduced tumor growth and increased lifespans in animals treated with targeted nanogels.	[58]
AMT	TAT	TAT-AT7-modified PEI nanocomplex	Secretory endostatin gene	-	Intravenous injection	Orthotopic U87 glioma-bearing nude mice model	TAT-AT7 showed no obvious hemolysis at concentrations ranging from 2.5 to 640 µmol/L, indicating good biosafety.	The cellular uptake of TAT-AT7 in bEnd.3 cells (mouse brain microvascular endothelial cells) and its distribution in an intracranial glioma model were evaluated, demonstrating a high uptake efficiency and penetration capability.	TAT-AT7 exhibited significant inhibitory effects on HUVEC proliferation, migration, invasion, and tubular structure formation and also promoted apoptosis in HUVECs and inhibited zebrafish embryo angiogenesis. In vivo, TAT-AT7 significantly suppressed glioma growth, induced glioma cell apoptosis, and inhibited angiogenesis.	[65]
CMT+ RMT	RF + active targeting	LMP tFNA NPs	LMP	-	Tail vein injection	Orthotopic U87 glioma-bearing nude mice model	In vitro and in vivo safety evaluations showed that RFA NPs had no significant cytotoxic effects on primary hepatocytes, astrocytes, or GBM cells, and no obvious immune side effects were observed in vivo.	Biomimetic NPs encapsulated with natural cell membranes prolonged the circulation time of the drug in vivo. The hybrid membrane coating facilitated the efficient crossing of the BBB.	Inhibition of GBM growth: LMP-loaded RFA NPs exhibited superior and specific anti-GBM activities in vitro and in vivo. LMP induced apoptosis and pyroptosis in GBM cells, reducing tumor growth. The RFA NPs demonstrated reduced off-target drug delivery, ensuring specificity.	[83]
CMT	Active targeting	Ang-RBCm@NM-(Dox/Lex)	Dox	Lexiscan (Lex)	Intravenous injection	Orthotopic U87MG human GBM tumor-bearing nude mice	The nanomedicine was evaluated for cytotoxicity using in vitro cell viability assays. While specific toxicity data are not detailed in the Abstract, in vivo studies assessed the therapeutic efficacy and survival outcomes, which indirectly reflect safety.	The nanomedicine demonstrated a prolonged blood circulation time, with an elimination half-life (t1/2,β) of 9.3 h for Ang-RBCm@NM-(Dox/Lex), which is longer than its RBCm@NM-(Dox/Lex) counterpart (t1/2,β = 7.8 h).	Superior BBB penetration: the angiopep-2 functionalization and Lex-mediated BBB opening facilitated superb penetration across the BBB Treatment with Ang-RBCm@NM-(Dox/Lex) resulted in effective tumor growth suppression and significantly improved median survival time in orthotopic U87MG human GBM tumor-bearing nude mice.	[81]
CMT	Active targeting	iRGD-EM:TNDs	TMZ	-	Tail vein injection	Orthotopic U87MG human GBM tumor-bearing nude mice	Safety was evaluated through the hematoxylin and eosin (H&E) pathological staining of major organs, the monitoring of body weight, and the detection of IgE levels associated with hypersensitivity reactions. The results suggest the low systemic toxicity and good biocompatibility of the iRGD-EM:TNDs.	The EM-coated nanodots demonstrated a longer elimination half-life, suggesting reduced degradation in vivo compared to traditional PEG stealth motifs.	iRGD-EM:TNDs showed enhanced cellular uptake, improved penetration in multicellular tumor spheroids, and increased transport ratios across the BBB in vitro and in vivo. The treatment with iRGD-EM:TNDs resulted in a 100% survival rate after 30 days post-tumor implantation and induced the highest cell apoptosis level.	[82]
CMT	IL13Rα2	T cells + BPLP-PLA-NPs (clicked)	DOX	CAR T-cell therapy	Tail vein injection	Intracranial xenograft model using female immunodeficient nude mice	The safety evaluation included assessing the cytotoxic effects of the nanoparticles and T cells on GBM cells in vitro and observing the behavior of T cells in vivo without them causing observable side effects.	The study evaluated the retention of nanoparticles on T cells for at least 8 days, indicating the stability of the linkage for a suitable time window for in vivo delivery.	The system demonstrated enhanced cytotoxic effects in vitro with T cells clicked with doxorubicin-loaded nanoparticles compared to bare T cells. In vivo, T cells expressing TQM-13 served as delivery shuttles for nanoparticles, significantly increasing the number of nanoparticles reaching brain tumors compared to nanoparticles alone.	[88]
CMT	Inflammatory tumor microenvironment	NEs-Exos/DOX	DOX	-	Intravenous injection	C6-Luc glioma-bearing mice models	The safety evaluation of the NEs-Exos/DOX system is not detailed in the provided text. The focus is on the efficacy of the system in crossing the BBB and targeting glioma cells.	The study does not provide specific pharmacokinetic data for the NEs-Exos/DOX system. However, it does mention that NEs-Exos can rapidly penetrate the BBB and migrate into the brain, suggesting favorable pharmacokinetic properties for brain tumor targeting.	The pharmacodynamics of the NEs-Exos/DOX system were demonstrated through in vitro and in vivo assays, showing that the system can improve the anticancer efficacy of DOX, reduce mortality, and effectively suppress tumor growth while prolonging the survival time in a glioma mouse model.	[89]
CMT	Domain CLTX and IgG4 hinge	CAR-neutrophils@RSiO2-TPZ	Tirapazamine (TPZ)	-	Intravenous injection	Mouse xenograft model of GBM	CAR neutrophils exhibited high biocompatibility with normal cells (SVG p12 glial cells, hPSCs, and hPSC-derived cells). Necrosis was not observed in major organs of experimental mice. However, concerns regarding off-target tissue toxicity or systemic toxicity are mentioned.	CAR neutrophils delivered >20% of administered nanodrugs to brain tumors compared to 1% via free nanodrugs.	CAR neutrophils presented enhanced antitumor cytotoxicity compared to peripheral blood (PB) neutrophils. In vivo tumor growth inhibition: CAR-neutrophils loaded with TPZ-loaded SiO_2_ nanoparticles significantly inhibited tumor growth and prolonged animal survival in GBM xenograft models. Mechanism: combination of CAR-enhanced direct cytolysis and chemotherapeutic-mediated tumor killing via cellular uptake and glutathione (GSH)-induced degradation of nanoparticles within tumor cells.	[90]
CMT	Active targeting	MPM@P NGs	Cisplatin	Chemodynamic therapy (CDT)	Intravenous injection	Orthotopic C6 glioma in a mouse model	The nanogel safety was evaluated using in vitro hemolysis assays, in vivo hematological indices, blood biochemical analysis, and a histopathological examination of major organs.	The nanogel PK was assessed by tracking the platinum (Pt) content in blood after intravenous injection. The MPM@P NG half-life was determined to be longer than that of PM@P NGs without a membrane coating, indicating an improved blood circulation time.	The in vivo antitumor activity of the nanogels was evaluated by monitoring tumor volume changes using T1-weighted MR imaging. The MPM@P NGs demonstrated the smallest tumor size and most efficient therapeutic effect among all the groups due to the combination of enhanced CDT and chemotherapy, as well as an improved BBB-crossing and glioma-targeting ability.	[75]
CMT	Active targeting	aDCM@PLGA/RAPA	RAPA	Immunotherapeutic activation	Tail vein injection	Orthotopic C6 tumor model	Studies investigating the biocompatibility and potential toxicity of aDCM@PLGA/RAPA in vivo and in vitro are mentioned but specific details are not provided.	Stability studies on aDCM@PLGA/RAPA demonstrated good colloidal stability in PBS and plasma, suggesting the potential for a prolonged circulation time in the blood.	aDCM@PLGA/RAPA effectively activated T cells and NK cells, modifying the tumor microenvironment to an immune-supportive state. Antitumor efficacy: the combined immunotherapeutic and chemotherapeutic effects led to the significant inhibition of glioma growth and induced glial differentiation.	[94]
CMT	Active targeting	Nanoparticle-engineered TRAIL-expressing hADSCs	TRAIL	-	Local intracranial delivery	Mouse intracranial xenograft model of patient-derived GBM cells	MTS assay revealed no significant change in cell viability of hADSCs transfected with nanoparticle-laden TRAIL DNA compared to controls, indicating safety. The in vivo safety was assessed by monitoring the survival and weight changes of the mice.	The TRAIL-overexpressing hADSC PK was evaluated by observing the migration and infiltration of the cells towards GBM tumors in vivo and measuring the TRAIL protein expression levels in vivo.	The study demonstrated that TRAIL-overexpressing hADSCs induce significant apoptosis in GBM cells in vitro and in vivo, with negligible apoptotic activity in normal brain cells. The therapeutic effects included tumor growth inhibition, the extension of animal survival, and reductions in the tumor mass and microsatellite occurrence.	[97]
CMT	Active targeting	PEI-PLL-transfected MSCs	Suicidal genes, namely, HSV-TK and TRAIL	Ganciclovir	Intratumoral injection	SD rats used with C6 glioma cells injected intracranially	Cell viability of MSCs transfected with the PEI-PLL copolymer was evaluated using the MTT assay. Cell viability was more than 90% at a pDNA/polymer ratio of 1:1.5 for the PEI-PLL copolymer. In vivo toxicity: while not explicitly stated in the Abstract or Methods sections, the use of nonviral vectors such as PEI-PLL copolymers is generally considered safer than viral vectors due to their low immunogenicity and reduced oncogenicity risk.	The PK of the system was assessed by monitoring the survival rates and tumor growth in the animal model after treatment with PEI-PLL-transfected MSCs.	The study demonstrated that the combination of HSV-TK and TRAIL genes in MSCs leads to a significant decrease in cell viability and an increase in apoptosis in glioma cells, both in vitro and in vivo. The reduction in the cell proliferation marker Ki67 and angiogenesis marker VEGF, along with the TUNEL assay results, indicate the therapeutic effectiveness of the MSCs at inducing apoptosis in GBM cells.	[99]
CMT	Active targeting	ApoE-ARTPC@TMZ	ART	TMZ	Tail vein injection	Orthotopic U251-TR GBM mouse model	In vivo experiments were conducted to assess the nanoplatform safety, including an examination of body weights, blood cell counts (RBCs, WBCs, PLTs), and histological examinations of major organs and brain tissue.	The circulation time of the liposomes in the bloodstream was evaluated using FITC-Dex as a marker, showing prolonged circulation times for ApoE-ARTPC@FITC-Dex and ARTPC@FITC-Dex compared to those of free FITC-Dex.	The induction of apoptosis, DNA damage, and inhibition of MGMT expression and Wnt/β-catenin signaling were assessed. The combination therapy showed an enhanced cytotoxicity, increased ROS generation, and significant apoptosis induction in vivo.	[106]

**Table 2 cancers-16-03300-t002:** Clinical trials of glioma treatment using NDDSs: ongoing and completed.

Methods to Overcome the BBB	Drug	Combination Therapy (If Any)	Research Phases	NDDSs	Administration Route	Safety Evaluation	PK	Primary Efficacy Endpoints	ClinicalTrials.Gov Identifier	Ref.
Passive diffusion	DOX	TMZ + radiotherapy	Phase II	Caelyx™, PEG-Dox	Intravenous infusion + oral administration	The treatment was well tolerated, with most AEs classified as grade 1–2. However, some grade 3–4 AEs were also reported.	The PK of PEG-Dox and temozolomide were not specifically detailed in the document summary provided. However, the improved BBB penetration of PEG-Dox suggests an enhanced PK compared to conventional doxorubicin.	Progression-free survival after intravenous infusion at 12 months (PFS-12): 30.2% in all patients. Median overall survival (mOS): 17.6 months in all patients including those from Phase I. Comparison to historical control: the endpoints did not differ significantly from the EORTC26981/NCIC-CE.3 data in a post hoc statistical comparison.	NCT00944801	[128]
Intratumoral injection	Magnetic iron oxide nanoparticles	Fractionated stereotactic radiotherapy	Phase II	Magnetic iron oxide nanoparticles	Intratumoral injection	Side effects: acute side effects during thermotherapy were classified according to the Common Toxicity Criteria (CTC) version 2.0. Common side effects included sweating (50.0%), a general sensation of warmth (47.0%), and thermal stress with body temperatures exceeding 38° C in 6 patients (9.1%). No serious complications were observed.	The study does not provide specific pharmacokinetic data. However, it mentions that no indication of iron release from the intratumoral deposits or iron metabolization was observed, suggesting that the nanoparticles remained stable post-administration.	Overall survival after first tumor recurrence (OS-2): the median OS-2 was 13.4 months (95% CI: 10.6–16.2 months) among the 59 patients with recurrent glioblastoma. Only the tumor volume at study entry was significantly correlated with ensuing survival (*p* < 0.01). Overall survival after primary tumor diagnosis (OS-1): the median OS-1 was 23.2 months, with a 95% confidence interval of 17.2–29.2 months.	-	[129]
EPR	Irinotecan	-	Phase I	nal-IRI	Intravenous infusion	The maximum tolerated dose (MTD) was determined for both WT and HT cohorts. Dose-limiting toxicities included diarrhea, dehydration, and fatigue. The study concluded that nal-IRI had no unexpected toxicities and that the UGT1A1 genotype did not correlate with toxicity.	PK results were comparable to those seen in other PK studies of nal-IRI. PK parameters were analyzed, including maximum plasma concentrations (Cmax), areas under plasma concentration–time curve (AUC0−t), and terminal half-life (t1/2) for total irinotecan, SN-38, and SN-38G. UGT1A1*28 genotype did not affect PK parameters.	The primary efficacy endpoint was PFS-6. The study reported PFS-6 as 2.9% for the intent-to-treat cohort, with a median PFS of 42 days and a median overall survival of 107 days.	-	[127]
Passive diffusion	Irinotecan	TMZ	Phase I	nal-IRI	Intravenous infusion + oral administration	The study evaluated safety by monitoring dose-limiting toxicities (DLTs), which included grade 4 neutropenia, grade 3 diarrhea, hypokalemia, fatigue, anorexia, and other grade 3 or 4 nonhematologic toxicities. The MTD for nal-IRI was determined to be 50 mg/m^2^ every 2 weeks with TMZ.	The study does not provide specific pharmacokinetic data. Enhanced BBB penetration: the nanoliposomal encapsulation of irinotecan improved its ability to cross the BBB, as demonstrated in preclinical animal models. Tissue analysis: in preclinical studies, nal-IRI showed a 10.9-fold increase in tumor area under the curve compared to free irinotecan and a 35-fold selectivity for tumor versus normal tissue exposure.	The primary efficacy endpoints were the assessment of the response rate (complete or partial response as defined by Macdonald criteria) and PFS. The study was terminated after an interim analysis showed no activity (0% response rate) and a median PFS of 2 months.	NCT03119064	[126]
EPR + active transport	NU-0129	-	Phase 0	siBcl2L12-SNAs	Intravenous infusion	The safety assessment revealed no significant treatment-related toxicities. The study monitored vital signs, blood chemistry, and adverse events in patients, with only two treatment-related severe adverse events (lymphopenia and hypophosphatemia) noted, which were considered “possibly” related to the treatment.	PK analysis showed that siRNA was rapidly eliminated from plasma with a mean half-life of 0.09 h, while gold (Au), used as a marker for the SNAs, had a much slower elimination with a mean half-life of 17 h. The study also details the clearance rates and volume of distribution for both siRNA and Au.	The primary efficacy endpoints were the intratumoral accumulation of SNAs and the suppression of the Bcl2L12 gene. The study reported that NU-0129 uptake into glioma cells correlated with a significant reduction in tumor-associated Bcl2L12 protein expression. Additionally, the presence of Au in the tumor tissue was confirmed, indicating that the SNAs reached the patient tumors.	NCT03020017	[130]
RMT	DOX	-	Phase I	Anti-EGFR ILs-dox	Intravenous infusion	The study reports safety data from the application of anti-EGFR ILs-dox in patients with relapsed glioblastoma. No grade 4 or 5 adverse events occurred. One case of severe pneumonitis was reported, which resolved with treatment. Other adverse events included febrile neutropenia in two patients, which was managed without sequelae.	The pharmacokinetic analysis showed that the mean plasma concentration of doxorubicin 24 h after administration was 15,805 ng/mL. DOX concentrations in CSF were below 1 ng/mL in all patients, indicating that anti-EGFR ILs-dox does not cross the BBB at clinically relevant levels. However, significant doxorubicin levels were detected in glioblastoma tissue 24 h after application, suggesting that the disrupted BBB in high-grade gliomas may enable liposome delivery into tumor tissue.	The primary efficacy endpoints were the anti-EGFR ILs-dox concentrations in plasma, CSF, and glioblastoma tissue. The median PFS was 1.5 months, and the median OS was 8 months. One patient had a very long remission, suggesting that neoadjuvant administration may positively affect the outcome.	NCT03603379	[131]
EPR	AGuIX nanoparticles	Standard of care for glioblastoma	Phase I/II	AGuIX nanoparticles	Intravenous infusion	Toxicity assessment: DLT defined as any grade 3–4 NCI Common Terminology Criteria for Adverse Events (CTCAE) toxicity, except for alopecia, nausea, vomiting, or fever. Adverse event reporting: according to CTCAE (version 5.0). Neurological status evaluation: clinical assessment and Mini-Mental State Examination (MMSE).	PK parameters of AGuIX nanoparticles, including AUC, Tmax, and Cmax, were measured on blood samples and urinary excretion during Phase I of the study.	For Phase I, the primary endpoint is the RP2D of AGuIX nanoparticles, with DLT defined as any grade 3–4 toxicity. For Phase II, the primary endpoint is the 6-month progression-free survival rate, which will be estimated using the Kaplan–Meier method.	NCT04881032	[132]
EPR + RMT	Mitoxantrone	-	Phase I	EGFR-Erbitux EDVsMIT	Intravenous infusion	The safety evaluation of EEDVsMit showed that it was well tolerated, with no dose-limiting toxicities observed. The most common drug-related adverse events were grade 1–2 fever, nausea, vomiting, rash, lymphopenia, and mildly deranged liver function tests.	The study does not provide detailed pharmacokinetic data within the summary. However, it mentions that the EDV selectively targets the cancer cell via the bispecific antibody and releases the cytotoxic drug into the tumor cell after macropinocytosis and a breakdown in lysosomes.	The primary efficacy endpoints were safety and tolerability, with the study also aiming to preliminarily define the antitumor activity of mitoxantrone-containing EDVs. The best antitumor response observed was a mixed response in one patient, and all other patients had progressive disease.	NCT02687386	[133]
CED	Irinotecan	-	Phase I	nal-IRI	Intratumoral injection	Safety was evaluated via monitoring adverse events, scheduled laboratory assessments, vital sign measurements, and physical examinations. Toxicities were graded according to the NCI CTCAE v. 4.0.	The study assesses the distribution of gadolinium, used in conjunction with real-time imaging, to model the drug distribution and evaluate the effectiveness of CED at delivering nal-IRI to the tumor site.	The primary efficacy endpoint is OS at 12 months (OS12), which will be considered evaluable for clinical efficacy.	NCT03086616	-
CED	Irinotecan	-	Phase I	Liposomal irinotecan	Intratumoral injection	Safety was evaluated by monitoring DLTs within 30 days post-infusion, as well as any neurological or systemic grade 3 or higher toxicities.	PK measurements were taken at pre-dose, 1 day after drug administration, and 1 week post-op to estimate the drug distribution and concentration in the brain.	The primary efficacy endpoints include the determination of the maximum tolerated dose and the assessment of the PFS at 6 months and 10 years, as well as the OS at 12 months and 10 years.	NCT 02022644	-
Active targeting	MT-201-GBM	Standard radiation therapy and TMZ chemotherapy	Phase I	MT-201-GBM	Intravenous administration	The study aimed to assess the safety and tolerability of MT-201-GBM by measuring the primary safety endpoint: the percentage of patients with a DLT during the DLT observation period within each dose level. Other safety measures: changes in laboratory parameters (e.g., hemoglobin, creatinine, AST, and bilirubin) and monitoring for adverse events.	The study does not provide specific details on pharmacokinetic assessments. However, it does include immune response measurements, such as changes from baseline in IFN gamma, granzyme-B, and fluorospot, which may provide insights into the vaccine’s biological activity.	The primary efficacy endpoints include the MTD of MT-201-GBM and immune response measurements after the second and third infusions compared to baseline. Secondary outcome measures include OS, PFS, and changes in immune response indicators.	NCT04741984	-
Active targeting and bypassing of the BBB	5-FC	-	Phase I	CD-NSCs	Intracranial injection + oral administration	The study evaluated safety through the monitoring of adverse events, an assessment of possible NSC migration into the systemic circulation, and testing for the presence of replication-competent retrovirus (RCR). No DLTs were observed due to the CD-NSCs, and there was no development of anti-CD-NSC antibodies.	Intracerebral microdialysis was used to measure brain levels of 5-FC and 5-FU, showing CD-NSCs produced 5-FU locally in a 5-FC dose-dependent manner. Blood samples were collected to assess systemic drug concentrations and perform immunologic correlative studies.	The primary efficacy endpoints included assessing the feasibility of treating recurrent high-grade glioma patients with intracranially administered CD-NSCs followed by oral 5-FC. Secondary objectives included assessing possible CDNSC immunogenicity and secondary tumorigenicity and evaluating the proof-of-concept regarding CD-NSC migration to tumor foci and the localized conversion of 5-FC to 5-FU.	NCT01172964	[134]
Active targeting and bypassing of the BBB	5-FC	Leucovorin	Phase I	CD-NSCs	Intracranial injection	Safety was evaluated by monitoring adverse events, an assessment of possible NSC migration into the systemic circulation using qPCR, and testing for the presence of replication-competent retrovirus (RCR). No clinical signs of immunogenicity were observed, and only three patients developed anti-NSC antibodies.	Neuropharmacokinetic data were collected using intracerebral microdialysis to confirm continuous production of 5-FU in the brain during the course of 5-FC administration. The primary pharmacokinetic parameters of interest were the Cmax and AUC of 5-FC and 5-FU, measured in both dialysate samples and plasma.	The primary efficacy endpoints included assessing the feasibility of serially administering CD-NSCs and determining the recommended dosing for Phase II testing. Secondary objectives included characterizing the relationship between intracerebral and systemic concentrations of 5-FC and 5-FU, assessing CD-NSC immunogenicity, and describing clinical activity. The best response observed was stable disease in three participants, with a range of 4 to 5 months.	NCT02015819	[135]
Active targeting and bypassing of the BBB	Irinotecan Hydrochloride	-	Phase I	hCE1m6-NSCs	Intracranial administration + intravenous administration	Safety was evaluated by grading toxicities using the NCI CTCAE version 4.0, with specific attention to DLTs during the first treatment cycle.	PK data were collected via intracerebral microdialysis to measure SN-38 concentrations in the brain interstitium and plasma levels of irinotecan and SN-38.	The primary efficacy endpoints included the assessment of clinical benefit, defined as stable disease, partial response, or complete response, as measured by RANO criteria, for up to 6 months.	NCT02192359	-
Active targeting and bypassing of the BBB	Oncolytic Adenovirus (CRAd-S-pk7)	Standard temozolomide chemotherapy and radiotherapy	Phase I	NSC-CRAd-S-pk7	Intracranial injection	The trial demonstrated that the treatment was safe, with no formal dose-limiting toxicity reached. One patient developed viral meningitis (grade 3) due to an inadvertent injection into the lateral ventricle.	PK assessments were performed through medical history, imaging, blood samples, and chemistry values collected at screening and follow-up. The study does not detail specific pharmacokinetic parameters but focuses on the safety and presence of the treatment.	The primary endpoints were safety and tolerability. Secondary endpoints included the estimation of survival outcomes and evaluation of immune response correlation with clinical outcomes. The median progression-free survival was 9.05 months, and the overall survival was 18.4 months.	NCT03072134	[136]
CMT	Survivin DC cell injection	Radiotherapy and TMZ chemotherapy	Phase I	Survivin-loaded dendritic cell injection	Intradermal injection + intravenous infusion	Safety was evaluated by monitoring adverse events that occurred within 28 days after the first to the third dose, as judged by the investigator to be related to the study drug. DLT was a primary outcome measure.	Not specifically detailed in the document. However, the administration schedule (days 0, 14, and 28) and route of administration (ID and IV) indicate that the PK is influenced by the absorption, distribution, metabolism, and excretion of the survivin-loaded dendritic cells.	The primary efficacy endpoints include the PFS and OS measured over 2 years. Secondary outcome measures include immune effect assessments such as an anti-survivin antibody test, cytokine detection, and specific T-cell responses, as well as DC cell activity and in vivo processes, measured at specific time points.	NCT06524063	-

## Data Availability

No new data were created or analyzed in this study. Data sharing is not applicable to this article.

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
