# Peer review of "Blood–Brain Barrier Conquest in Glioblastoma Nanomedicine: Strategies, Clinical Advances, and Emerging Challenges"

_cancers, 2024, doi:10.3390/cancers16193300_

Round 1
Reviewer 1 Report
Comments and Suggestions for Authors
Review of Blood-Brain Barrier Conquest in Glioblastoma Nanomedicine: 2 Strategies, Clinical Advances, and Emerging Challenges (Manuscript ID: cancers-3213967)
The authors Duan et. al. have presented their findings on the application of NDDS in the treatment of GBM with special emphasis on the clinical trials and strategies to make NDDS more successful in the same. The review is very thorough and presents a lot of useful information. However, I have the following comments for the authors:
Line 55 “Due to its heterogeneity,……” is incorrect. Please correct the language and grammar.
The definitions of RMT, CMT, TMT, etc mentioned in the Table 1 under the strategies to cross BBB should be given prior to the table or as a legend for the benefit of the readers.
What is the difference between section 2 and 3. They both seem to have the same title. Section 3 should be “Alternative strategies for NDDS to bypass the BBB”.
Comments on the Quality of English LanguagePlease check the language and grammar overall. There are inconsistencies.
Reviewer 2 Report
Comments and Suggestions for Authors
Glioblastoma is a common malignancy in the central nervous system and has poor chances of recovery. Standard glioblastoma treatment involves surgical tumor removal, radiotherapy, and chemotherapy, but overall outcomes are still unsatisfying. The blood-brain barrier hinders the effective delivery of therapeutic agents to glioblastoma despite its role in CNS tissue stability. Recent preclinical studies show nanomedicine delivery systems (NDDS) can effectively target and safely treat GBM, offering the potential for targeted drug delivery. This review discussed strategies utilized in preclinical studies to overcome the blood-brain barrier for drug delivery and provided an overview of the progress in the clinical translation of novel drug delivery systems. The discussion also included potential strategies for advancing the development of NDDS and expediting translational research through well-designed clinical trials for glioblastoma multiforme therapy. Overall, this manuscript is an extensive review work with well-organized figures and tables. I would recommend its publication if the authors could address the following minor issues:
1. Focused ultrasound (FUS) should be mentioned in the caption of Figure 2.
2. In the first two paragraphs of the introduction, the blood-brain barrier (BBB) should be introduced first before the acronym is used.
3. Little introduction was made for Figure 4, especially in D. Please consider either removing this part or introducing the different treatments applied. The same applies to Figures 5 and 6.
Comments on the Quality of English LanguageMany acronyms were listed before they were introduced. The authors should carefully check them.
